# Hydrogen sulfide (H₂S) coordinates redox balance, carbon metabolism, and mitochondrial bioenergetics to suppress SARS-CoV-2 infection

Ragini Agrawal[1,2,3], Virender Kumar Pal[1,2], Suhas K.S.[1,2], Gopika Jayan Menon[1,2], Inder Raj Singh[4], Nitish Malhotra[4], Naren C.S.[5], Kailash Ganesh[3], Raju S. Rajmani[6], Aswin Sai Narain Seshasayee[4], Nagasuma Chandra[5], Manjunath B. Joshi[3]*, Amit Singh [1,2]*

1 Department of Microbiology and Cell Biology, Indian Institute of Science, Bengaluru, Karnataka, India, 2 Centre for Infectious Disease Research, Indian Institute of Science, Bengaluru, Karnataka, India, 3 Department of Aging Research, Manipal School of Life Sciences, Manipal Academy of Higher Education, Udupi, Karnataka, India, 4 National Centre for Biological Sciences, Tata Institute of Fundamental Research, Bengaluru, Karnataka, India, 5 Department of Biochemistry, Indian Institute of Science, Bengaluru, Karnataka, India, 6 Molecular Biophysics Unit, Indian Institute of Science, Bengaluru, Karnataka, India

* manjunath.joshi@manipal.edu (MBJ); asingh@iisc.ac.in (AS)

## Abstract

Viruses modulate various aspects of host physiology, including carbon metabolism, redox balance, and mitochondrial bioenergetics to acquire the building blocks for replication and regulation of the immune response. Understanding how SARS-CoV-2 alters the host metabolism may lead to treatments for COVID-19. We report that a ubiquitous gaseous molecule, hydrogen sulfide (H₂S), regulates redox, metabolism, and mitochondrial bioenergetics to control SARS-CoV-2. Virus replication is associated with down-regulation of the H₂S-producing enzymes cystathionine-β-synthase (CBS), cystathionine-γ-lyase (CTH), and 3-mercaptopyruvate sulfurtransferase (3-MST) in multiple cell lines and nasopharyngeal swabs of symptomatic COVID-19 patients. Consequently, SARS-CoV-2-infected cells showed diminished endogenous H₂S levels and a protein modification (S-sulfhydration) caused by H₂S. Genetic silencing or chemical inhibition of CTH resulted in SARS-CoV-2 proliferation. Chemical supplementation of H₂S using a slow-releasing H₂S donor, GYY4137, diminished virus replication. Using a redox biosensor, metabolomics, transcriptomics, and XF-flux analyzer, we showed that GYY4137 blocked SARS-CoV-2 replication by inducing the Nrf2/Keap1 pathway, restoring redox balance and carbon metabolites and potentiating mitochondrial oxidative phosphorylation. Treatment of SARS-CoV-2-infected mice or hamsters with GYY4137 suppressed viral replication and ameliorated lung pathology. GYY4137 treatment reduced the expression of inflammatory cytokines and re-established the expression of Nrf2-dependent antioxidant genes in the lungs of SARS-CoV-2-infected mice. Notably, non-invasive measurement of respiratory functions using unrestrained whole-body plethysmography (uWBP) of

**Data availability statement:** All data needed to evaluate the conclusions in the paper are present in the paper and/or the Supplementary Materials. The RNA-seq datasets have been submitted in Gene Expression Omnibus (GEO) with accession number GSE283665. All the scripts for running the DGE and pathway analysis are publicly available at GitHUB (https://github.com/IRSINGH27/RNA-Seq-Eukaryote/tree/main) under the MIT license.

**Funding:** This work was supported by Scheme for Transformational and Advanced Research in Science (MoE-STARS/STAR-2/2023-0460), DST Supra Fund (SPR/2021/000175-G), Department of Biotechnology (BT/PR47905/MED/29/1643/2023) and Revati and Satya Nadham Atluri Chair Professorship to A.S. We are also grateful to the Crypto Relief Fund, L & T Trust (ODAA/INT/20-21), the DST-FIST program, the Institute of Eminence Fund, Ministry of Education, and the DBT-IISc partnership program to IISc. R.A. acknowledge fellowship from University Grant Commission (UGC). The funders had no role in study design, data collection and analysis, or preparation of the manuscript.

**Competing interests:** The authors have declared that no competing interests exist.

SARS-CoV-2-infected mice showed improved pulmonary function variables, including pulmonary obstruction (Penh), end-expiratory pause (EEP), and relaxation time (RT) upon GYY4137 treatment. Together, our findings significantly extend our understanding of $H_2S$-mediated regulation of viral infections and open new avenues for investigating the pathogenic mechanisms and therapeutic opportunities for coronavirus-associated disorders.

## Authors summary

Comprehensive knowledge of SARS-CoV-2 biology is needed to understand the mechanisms of SARS-CoV-2-induced disease and to develop strategies to control COVID-19. Several RNA viruses can modulate host redox homeostasis, carbon metabolism, and mitochondrial bioenergetics to enhance their replication through diverse mechanisms. Despite enormous efforts, the mechanisms and pathways explored by SARS-CoV-2 to support its replication within host cells are still largely unknown. We demonstrate that $H_2S$ gas participates in the modulation of SARS-CoV-2 infection by suppressing virus replication. SARS-CoV-2 infection decreased the expression of key proteins involved in $H_2S$ biogenesis, and the amount of $H_2S$ produced intracellularly. Interestingly, inhibition of $H_2S$ biogenesis promotes virus replication, and a pharmacological $H_2S$ donor (GYY4137) reduces SARS-CoV-2 replication and restores pulmonary function in animal models. Our systematic mechanistic dissection of the role of $H_2S$ in cellular bioenergetics, redox metabolism, and virus replication unifies many previous phenomena associated with various viral infections, including COVID-19.

## Introduction

Coronavirus 2019 disease (COVID-19), caused by the severe acute respiratory syndrome coronavirus 2 (SARS-CoV-2), emerged as a major global emergency with 776,798,873 confirmed cases of COVID-19 reported, including 7,074,400 deaths in 2020–2024 (https://data.who.int/dashboards/covid19/cases). While COVID-19 is now declared a non-concerning disease by world health organization (WHO), the continuous emergence of SARS-CoV-2 variants underscores the importance of mechanistic understanding of disease biology and the continuing need for effective therapies. One approach is to focus on host-generated gaseous signalling molecules, such as nitric oxide (NO), carbon monoxide (CO), and hydrogen sulfide ($H_2S$). They modulate the immune response, inflammation, carbon metabolism, redox homeostasis, and bioenergetics of immune cells, thereby influencing viral infections [1–4]. For example, studies have revealed the antiviral role of NO and its beneficial effects in treating clinical complications of COVID-19 patients [5–7]. Additionally, the CO-producing enzyme hemeoxygenase-1 (HO-1) suppresses

SARS-CoV-2 replication by increasing interferons (IFNs) [8]. Furthermore, inducing HO-1 expression controls inflammation and coagulopathies in COVID-19 [8]. Despite the clinical evidence linking lower serum levels of $H_2S$ with increased COVID-19 disease severity and death [9], the systematic understanding of the antiviral effect of $H_2S$ remains circumstantial and poorly understood.

$H_2S$ is synthesized by diverse organisms, including mammals and bacteria, primarily via cystathionine-beta-synthase (CBS), cystathionine-gamma-lyase (CSE/CTH), and cysteine-aminotransferase (CAT)–3-mercaptopyruvate sulfurtransferase (3-MST). These enzymes are known to metabolize methionine and cysteine to form $H_2S$ [2,10]. Biochemically, $H_2S$ is a lipophilic molecule that rapidly crosses the cell membrane, dissociates into $HS^-$ and $S^{2-}$, and maintains an $HS^-$:$H_2S$ ratio of 3:1 at physiological pH [2,10]. Regulated production of $H_2S$ positively influences the function of immune cells by maintaining redox balance, stimulating mitochondrial bioenergetics, and reversing systemic inflammation; these $H_2S$-specific physiological changes are important for SARS-CoV-2 multiplication [10–12]. SARS-CoV-2 strongly inhibits host mitochondrial oxidative phosphorylation (OXPHOS), resulting in an increase in production of mitochondrial reactive oxygen species (mROS) [13]. Importantly, mROS stabilizes HIF-1α, which redirects the flow of carbon metabolites away from the TCA cycle through glycolysis/Pentose phosphate pathway (PPP), to provide substrates for viral multiplication [13].

We expect that the beneficial effect of $H_2S$ mitigation of oxidative stress and revamping of mitochondrial OXPHOS counteracts viral biogenesis and propagation. Indeed, $H_2S$ significantly attenuates replication of several respiratory viruses and virus-induced inflammation [2,14]. It has also been suggested that $H_2S$ interferes with SARS-CoV-2 entry into host cells by modulating the expression of angiotensin-converting enzyme 2 (ACE2) and transmembrane protease serine 2 (TMPRSS2) [15,16]. Additionally, perturbations in endogenous $H_2S$ levels are linked to cardiac disorder, metabolic syndrome, and lung failure, each of which is a risk factor for developing severe forms of COVID-19. Moreover, inhalation of $H_2S$ reduces symptoms and promotes recovery in COVID-19 patients [17]. Together, these results suggest that $H_2S$ is likely to counteract the pathogenesis of SARS-CoV-2. However, the mechanistic underpinnings of $H_2S$-mediated regulation of SARS-CoV-2 replication and therapeutic potential of slow-releasing $H_2S$ compounds for SARS-CoV-2 infection have yet to be explored.

We hypothesize that intracellular levels of $H_2S$ modulate SARS-CoV-2 infection by regulating redox balance, mitochondrial bioenergetics, and inflammation. To test this idea, the present work examined SARS-CoV-2 infection using *in-vitro* (cell line-based) and *in-vivo* (mice and hamsters) infection models. We also used biochemical and genetic approaches to investigate the functional link between $H_2S$ and SARS-CoV-2 replication. Finally, we used RNA-sequencing, untargeted metabolomics, and real-time extracellular flux analysis to examine the role of $H_2S$ in mediating SARS-CoV-2 replication by regulating gene expression, redox metabolism, and mitochondrial bioenergetics.

## Results

### Diminished biogenesis of endogenous $H_2S$ during SARS-CoV-2 infection

To examine the association between SARS-CoV-2 infection and $H_2S$, we measured the expression of *cbs, cth,* and *mst*, genes involved in endogenous $H_2S$ production [2,18](Fig 1A). We infected Vero C1008 clone E6 (VeroE6) cells, Angiotensin-converting enzyme 2-expressing HEK293T (HEK-ACE2) cells, and cultured human airway epithelial cells (Calu-3) cell lines with two SARS-CoV-2 strains (Hong Kong/VM20001061/2020 [HK] and B. 1.617. 2 [Delta]) and measured expression using RT-qPCR. First, we confirmed virus proliferation by measuring SARS-CoV-2 nucleocapsid (*n*) gene transcript. Virus-infected cell lines uniformly showed time- and MOI (multiplicity of infection)-dependent increase in the expression of *n* gene (Fig 1B). Infected VeroE6 cells also exhibited a time- and MOI-dependent decrease in mRNA expression from *cbs* and *mst* (Fig 1C and 1E). The *cth* transcript remained below the detection level using RT-qPCR (> 32 Ct value), but RNA-sequence data, described later, confirmed down-regulation of *cth* (Fig 1D). We did not observe difference in the magnitude of downregulation induced by HK compared to Delta variants (Fig 1C and 1E). As with VeroE6, infection of Calu-3 with SARS-CoV-2-HK reduced expression of *cbs* and *mst* at each time point tested (S1A Fig).

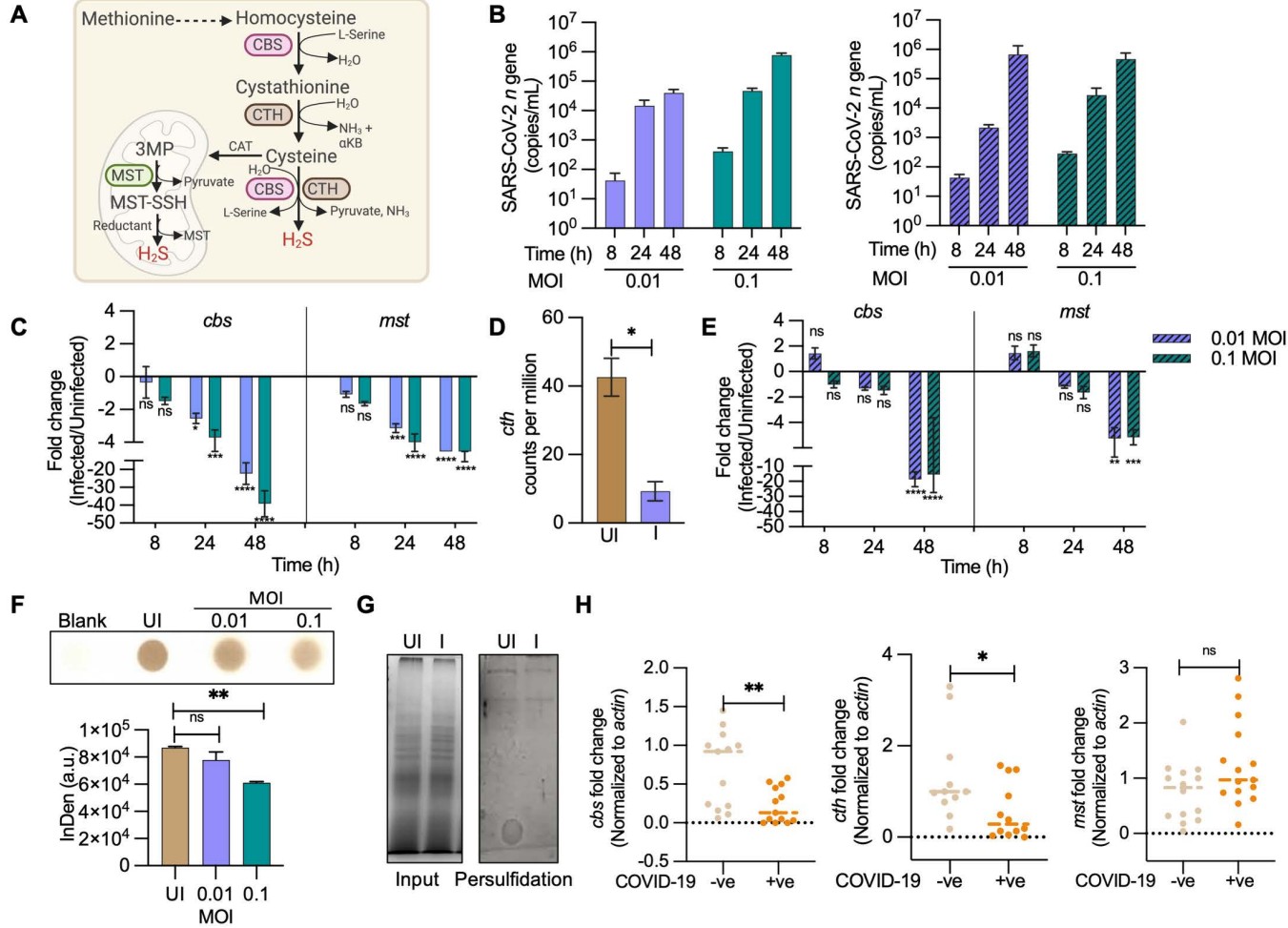

**Fig 1. Diminished biogenesis of endogenous H₂S during SARS-CoV-2 infection:** (A) Schematic showing H₂S producing enzymes in mammalian cells. (B) VeroE6 cells were infected with SARS-CoV-2 and *n* gene transcript was measured at the indicated times; solid bars = SARS-CoV-2-HK, hatched bars = SARS-CoV-2-Delta. (C) Time-dependent changes in expression of *cbs* and *mst* during SARS-CoV-2-HK replication in VeroE6 cells. (D) Transcript counts of *cth* upon SARS-CoV-2-HK infection in VeroE6 cells at 48 h pi by RNA-seq. (E) Time-dependent changes in expression of *cbs* and *mst* during SARS-CoV-2-Delta replication in VeroE6 cells. (F) VeroE6 cells were infected with SARS-CoV-2-HK. Endogenous H₂S levels were measured at 48 h pi by lead acetate assay (InDen- Integrated density). (G) Total cellular persulfide levels as measured by the ProPerDP method (UI- Uninfected; I- Infected). (H) *cth*, *cbs* and *mst* gene expression by RT-qPCR from nasopharyngeal swab samples of COVID-19 positive and negative individuals. Each dot represents an individual patient sample (n > 10). Results are expressed as mean ± standard deviation; representative of data from two independent experiments, performed thrice. *, $p < 0.05$; **, $p < 0.01$; ***; $p < 0.001$, ****, $p < 0.0001$, ns = non-significant either by two-way ANOVA with Dunnett's multiple comparison test (C,E) or student's t test with Welch's correction (D,F,H). (See also S1 Fig). Fig 1A graphic created in Biorender https://BioRender.com/ttos191.

Expression of *cth* was down-regulated only at 48 h pi and was followed by a recovery response at 72 h pi (S1A Fig). With HEK-ACE2 cells, SARS-CoV-2-HK infection uniformly suppressed expression of *cbs, cth*, and *mst* (S1B Fig). Consistent with the transcript data, CBS, CTH, and MST protein levels were down-regulated during SARS-CoV-2-HK infection of HEK-ACE2 cells (S1C Fig), indicating reduced H₂S biogenesis during infection.

We next measured cellular generation of H₂S using a lead acetate assay [19]. As with expression data, infection by SARS-CoV-2-HK reduced H₂S levels in lysates of VeroE6 cells (Fig 1F). Since H₂S mediates its effect through S-persulfidation (RSSH) of specific cysteine residues on proteins [20], we assessed the cellular levels of S-persulfidation

using a protein persulfide detection protocol (ProPerDp) [21]. SARS-CoV-2-HK-infected VeroE6 cells showed a marginal but uniform reduction in cellular S-persulfidated proteins relative to uninfected cells (Fig 1G), thus reinforcing the idea that virus infection reduces $H_2S$ biogenesis.

We also examined the possible clinical relevance of the gene expression findings by measuring expression of *cth*, *cbs*, and *mst* in the RNA derived from nasopharyngeal swabs of symptomatic COVID-19 patients. A significant decrease in the expression of *cbs* and *cth* was observed relative to samples from patients uninfected with SARS-CoV-2 but displaying respiratory distress due to other complications (Fig 1H). Overall, the data indicate that SARS-CoV-2 infection is associated with diminished biogenesis of endogenous $H_2S$.

## $H_2S$ suppresses SARS-CoV-2 proliferation

Since a reduction in endogenous $H_2S$ is associated with SARS-CoV-2 replication, we asked whether perturbation of endogenous $H_2S$ levels can regulate SARS-CoV-2 proliferation (Fig 2A). When we depleted endogenous CTH levels in VeroE6 cells using RNA interference (RNAi), the short hairpin RNA (shRNA) specific for CTH (Vero-shCTH) silenced expression of CTH by ~60% relative to results with scrambled shRNA (Vero-shScr) (Fig 2B). We infected Vero-shCTH and Vero-shSCR with SARS-CoV-2-HK and investigated the effect of CTH suppression by measuring *n* gene transcript in the cell supernatant at 8 and 24 h post-infection (p.i). We found that expression of the SARS-CoV-2 *n* gene in Vero-shCTH increased by 14- and 10-fold at 8 h ($p = 0.057$) and 24 h p.i ($p = 0.009$), respectively, compared to Vero-shScr (Fig 2B). As an additional test, we suppressed endogenous $H_2S$ production by treating VeroE6 cells with D,L-propargylglycine (PAG), a widely used selective inhibitor of CTH (Fig 2A) [22]. We found a uniform increase in *n* gene expression, which indicated elevated virus multiplication at 8 and 24 h post-treatment (Fig 2C). We also reduced expression of CBS and MST by RNAi (S2A and S2B Fig) and assessed the expression of virus transcript. The reduction of MST marginally increased virus multiplication, whereas CBS silencing either did not influence or marginally reduced virus replication (S2C Fig).

Having shown that diminished levels of endogenous $H_2S$ are associated with virus proliferation, we next asked whether elevating $H_2S$ levels, using the small-molecule $H_2S$ donors P-(4- Methoxyphenyl)-P-4-morpholinyl-phosphinodithioic (GYY4137) and GYY4137 sodium salt (Na-GYY4137) (Fig 2A), suppresses SARS-CoV-2. GYY4137/Na-GYY4137 are established $H_2S$ donors that generate a low, sustained amount of $H_2S$ over a prolonged period, to mimic endogenous $H_2S$ production [23,24]. Na-GYY4137 is devoid of potential off-target effects associated with the use of dichloromethane-complexed GYY4137 morpholine salt form. Since $H_2S$ release is slow by these donors, the final concentration of $H_2S$ released is usually much lower than the initial concentration of GYY4137. We confirmed this by quantifying $H_2S$ release in VeroE6 cells treated with 5 mM of GYY4137 and Na-GYY4137 using a methylene blue colorimetric assay [25]. As a control, we exposed VeroE6 cells to 5 mM NaHS, which rapidly releases a high amount of $H_2S$ [26] and has been recently shown to be ineffective in suppressing SARS-CoV-2 replication [27]. Expectedly, NaHS completely released $H_2S$ within 15 min of treatment (Fig 2D). In contrast, GYY4137 donors consistently released ~600–700 μM of $H_2S$ for the entire 48-h duration of the experiment (Fig 2D). Cytotoxicity assays revealed that 5 mM of GYY4137 donors did not elicit lethality in VeroE6 cells (S2D Fig).

We systematically examined the effect of GYY4137 donors on SARS-CoV-2 infection using multiple models. VeroE6, HEK-ACE2 and Calu-3 cells, pre-treated with 5 mM of GYY4137 or Na-GYY4137 for 4 h, were infected with SARS-CoV-2-HK for 1 h at an MOI of 0.01 or 0.1. At 48 h pi, viral load was measured in the cell supernatant by RT-qPCR and plaque assay. An ~100- to 1000-fold reduction in *n* gene expression was observed upon treatment with GYY4137 or Na-GYY4137 (Fig 2E). Consistent with this finding, a plaque assay showed a significant reduction in the titre of infectious virions upon treatment with GYY4137 or Na-GYY4137 (S2E Fig). As with the HK variant, GYY4137 uniformly suppressed multiplication of the Delta variant in VeroE6 and HEK-ACE2 cells (Fig 2E). As expected from RT-qPCR and plaque assay data, GYY4137 and Na-GYY4137 pre-treatment significantly protected VeroE6 and Calu-3 cells from virus-induced cell death, as evident in cell culture images (Fig 2F).

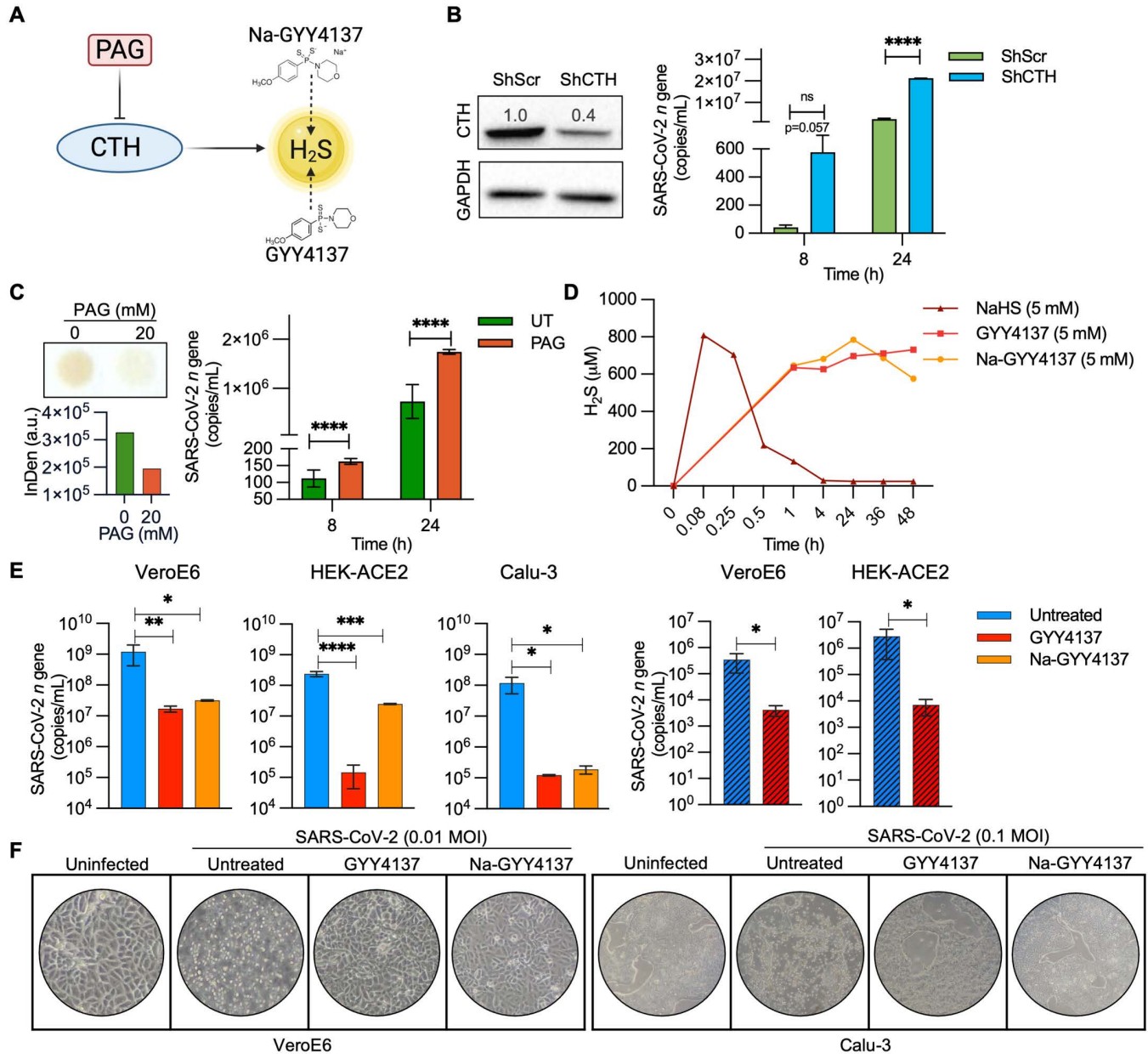

**Fig 2. H₂S suppresses SARS-CoV-2 proliferation:** (A) Schematic of the H₂S inhibitors and donors used in the study.(B) Genetic silencing of CTH was achieved in VeroE6 cells by lentiviral vectors, confirmed by western blotting and quantified by densitometric analysis using Image Lab software, followed by infection with 0.01 MOI of SARS-CoV-2-HK; virus replication was measured by RT-qPCR at indicated time intervals. (C) Endogenous H₂S production was suppressed by treating VeroE6 cells with 20 mM PAG, as confirmed by lead acetate assay, and subsequently infected with 0.01 MOI of SARS-CoV-2-HK; virus replication was measured by RT-qPCR at indicated time intervals (InDen- Intergrated density). (D) VeroE6 cells were treated with H₂S donors (NaHS, GYY4137 or Na-GYY4137) and media supernatant was harvested to assess H₂S production by methylene blue assay over time. (E) VeroE6, HEK-ACE2 and Calu-3 cells pre-treated with 5 mM GYY4137 or Na-GYY4137 for 4 h were infected with SARS-CoV-2. Virus replication was measured by RT-qPCR at 48 h pi; Solid bars = SARS-CoV-2-HK, hatched bars = SARS-CoV-2-Delta. (F) Representative culture well images of SARS-CoV-2-HK infected VeroE6 and Calu-3 cells in presence or absence of 5 mM GYY4137 or Na-GYY4137 at 48 h pi and 72 h pi, respectively. Results are expressed as mean ± standard deviation; representative of data from two independent experiments, performed thrice. *, $p < 0.05$; **, $p < 0.01$; ***, $p < 0.001$; ****, $p < 0.0001$, ns = non-significant, either by two-way ANOVA with Dunnett's multiple comparison test (B,C) or one-way ANOVA with Welch's correction (E). (See also S2 Fig). Fig 2A graphic created in Biorender, https://BioRender.com/zhpdqda.

Since we have pre-treated cells with the $H_2S$ donors, the suppression of virus multiplication could also be due to interference with the virus entry. Treatment with $H_2S$ donors is known to alter virus receptors such as ACE2 and TMPRSS2 expression in atherosclerosis, hypertension, and cancer [15,28,29]. We examined the expression of *ace2* and *tmprss2* in VeroE6 cells. As shown in S2F Fig, SARS-CoV-2 infection uniformly suppressed *ace2* expression in an MOI and time-dependent manner. However, treatment with GYY4137 did not influence *ace2* expression in the SARS-CoV-2 infected or uninfected VeroE6 cells (S2F Fig). TMPRSS2 expression is previously reported undetected in VeroE6 cells [30], which we confirmed by RT-qPCR assay in our experiments. Similarly, infection of Calu-3 cells resulted in a marginal down-regulation of *ace2* and *tmprss2*, which was not affected upon treatment with GYY4137 (S2G Fig). We further examined if GYY4137 affects virus entry, by first infecting VeroE6 cells with SARS-CoV-2 at a higher MOI of 3 for 1 h and then exposed to 5 mM of GYY4137. As a control, we pre-treated VeroE6 cells with GYY4137 and continued the treatment throughout the experiment. The expression of *n* gene was measured at 8 h post-infection (S2H Fig). Consistent with our earlier data, pre-treatment + continued treatment (Full time) resulted in ~ 300-fold reduction in *n* gene transcript compared to untreated samples. Although to a lesser extent, post-entry treatment with GYY4137 resulted in an ~ 40-fold decrease in *n* gene copy number than untreated cells. As an additional verification, we co-incubated VeroE6 cells with GYY4137 and SARS-CoV-2 virus for 1 h during the internalization phase. This was followed by washing both the virus and the $H_2S$ donor, and the measurement of *n* gene expression at 8 h p.i. An ~ 80-fold reduction in the *n* gene expression compared to untreated control was observed (S2H Fig). Therefore it appears that GYY4137 inhibits virus proliferation by targeting mechanisms associated with internalization and post-infection multiplication. Overall, these data establish that elevated levels of endogenous $H_2S$ suppress virus multiplication.

## GYY4137 modulates expression of the Nrf2/Keap1 regulon during SARS-CoV-2 infection

To dissect the mechanism of GYY4137-mediated suppression of SARS-CoV-2 proliferation, we performed RNA-sequencing (RNA-seq) of poly(A) RNA from infected VeroE6 cells with and without 5 mM of GYY4137 treatment at 48 h pi. The transcriptional response of SARS-CoV-2-HK-infected VeroE6 cells showed differential expression (DE) of ~8600 genes compared to uninfected VeroE6 cells (fold change [FC] >1.5; false discovery rate [FDR] <0.1) (Fig 3B). Principal component analysis (PCA) clearly distinguished the transcriptome of the infected sample from that of the uninfected control (Fig 3A). When we performed functional analysis using KEGG pathway enrichment, the host transcriptional response to SARS-CoV-2 infection separated the response into three major components: energy metabolism, redox metabolism, and cytokine-related transcriptional regulation centred around TNFα, RIG-1, and NF-κB signalling pathways (Fig 3C and 3D).

During infection with viruses, cellular detection of virus proliferation is mainly mediated by a family of intracellular pattern-recognition receptors (PRRs) that sense anomalous RNA structures generated during virus replication [31]. Among these, retinoic acid-inducible gene I (RIG-1)-like receptors (RLRs) are key sensors of virus infection. They mediate the activation of downstream transcription factors, most notably interferon regulator factors (IRFs) and NF-κB, which collectively establish the antiviral host response. Consistent with these principles, RNA-seq data showed enrichment of the RIG-1 receptor signalling pathway upon SARS-CoV-2 infection (Fig 3C). However, after sensing, pathways involved in the initial engagement of the antiviral response, such as type I interferon signalling (IFN-I) and interferon sensitive genes (ISGs), were either unaffected or down-regulated. This observation is consistent with the lack of IFN-I and inefficient IRF3 signalling in VeroE6 cells [32]. Genes associated with type III interferon signalling were marginally affected. In contrast, genes responsible for type II interferon production and regulation (*zc3h12a, ddit3, isg15, inhba, rara, rasgrp1, il18r1, hspd1, bcl3, abl1*) and cellular response (*ccl2, ifngr2, pim1, edn1, raf1, ptpn2, pparg, irf1, cdc42ep4, med1, stxbp4, dapk1, rpl13a, tlr2, zyx, actr3, nr1h2, pde12*), were induced. In the absence of type I and III IFN response, cells tend to respond to SARS-CoV-2 infection through type II IFN signalling in the VeroE6 cells. Despite aberrant type I/III IFN expression, the response to SARS-CoV2 infection in VeroE6 cells still elicited a strong chemotactic (*cd-147, il-6, ccr-2*) and

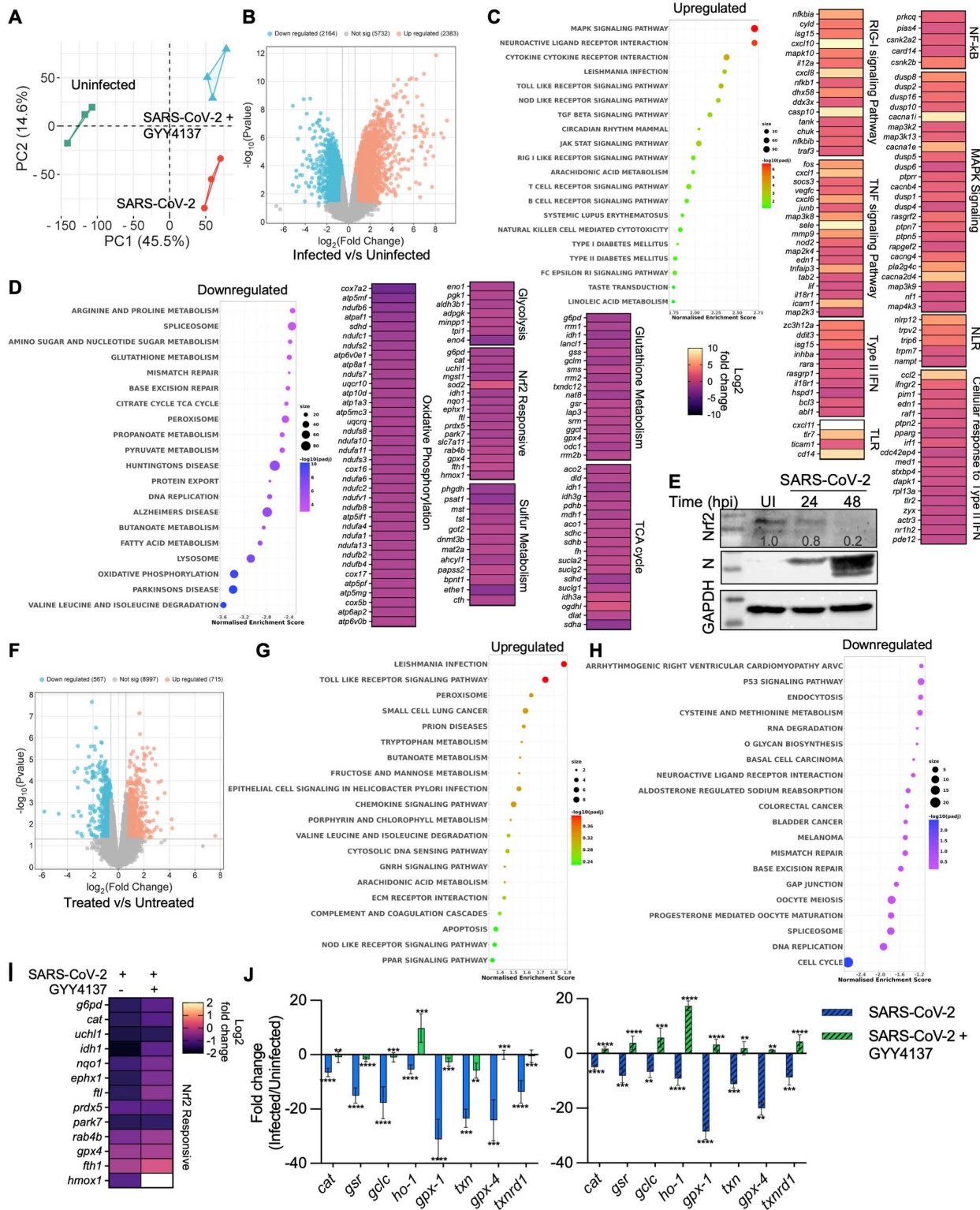

**Fig 3. GYY4137 modulates the expression of Nrf2/Keap-1 regulon during SARS-CoV-2 infection:** (A) PCA analysis plot of gene expression in uninfected and SARS-CoV-2-HK infection in the presence or absence of 5mM GYY4137. (B) Volcano plot visualizing differential expression between uninfected and infected samples. (C,D) KEGG terms mapping on differentially expressed genes for SARS-CoV-2-HK infection. Heat maps indicate the

fold change in infected cells with respect to uninfected cells. (E) Nrf2 levels were measured by immunoblotting from VeroE6 cells infected with SARS-CoV-2-HK, and quantified by densitometric analysis using Image Lab software. (F) Volcano plot visualizing differential expression between infected samples in presence and absence of 5 mM GYY4137. (G,H) KEGG terms mapping on differentially expressed genes for GYY4137 treatment upon SARS-CoV-2-HK infection. (I) Heat maps of Nrf2 responsive genes indicating the fold change in infected and GYY4137 treated cells with respect to uninfected cells. (J) RT-qPCR analysis of Nrf2 responsive genes from VeroE6 cells infected with SARS-CoV-2 (solid bars = SARS-CoV-2-HK, hatched bars = SARS-CoV-2-Delta) for 48 h, in the presence or absence of 5 mM GYY4137. Results are expressed as mean ± standard deviation; representative of data from two independent experiments, performed thrice. **, $p < 0.01$; ***, $p < 0.001$; ****, $p < 0.0001$, by one-way ANOVA with multiple comparison using Tukey's method (J). (See also S3 Fig).

proinflammatory response (*nfκb, tnf*). These findings agree with studies suggesting that reduced antiviral defence, coupled with inflammatory cytokine production, are critical features of COVID-19 [33].

SARS-CoV-2-associated NF-κB activation is also mediated by toll-like receptor (TLR) and TNFα signalling [34]. In agreement with these observations, we found increased expression of genes associated with TLR signalling (*cxcl11, tlr7, ticam1, and cd14*) and TNF-NF-κB signalling (*fos, cxcl1, socs3, cxcl6, mmp9, nod2, tnfaip3, prkcq, pias4, card14*) in SARS-CoV-2-HK-infected VeroE6 cells (Fig 3C). Activated NF-κB enters the nucleus and enhances the expression of downstream proinflammatory cytokines and chemokines [34]. As expected, SARS-CoV-2-HK infection induces significant upregulation of proinflammatory cytokines and chemokines in VeroE6 cells. NF-κB activation could be mediated by IL-6/JAK/STAT3 and mitogen-activated protein kinases (MAPKs) signalling [34,35]. All of these signalling components were induced in SARS-CoV-2-HK-infected VeroE6 cells (Fig 3C).

The induction of NF-κB/TLRs/TNF signalling pathways is associated with generation of reactive oxygen species (ROS), which contribute to acute lung injury in COVID-19 patients [36]. Consistent with this finding, we observed down-regulation of antioxidant genes (*e.g.,* glutathione peroxidases [*gpx4*], peroxiredoxins [*prdx5*], hemoxygenase (*hmox1*), catalase [*cat*], and thioredoxins [*txnrd1*]) regulated by nuclear factor erythroid 2-related factor 2 (Nrf2) – a central controller of cellular resistance to redox stress (Fig 3D) [37]. We validated RNA-seq data by showing that SARS-CoV-2 replication diminishes the levels of Nrf2 protein in Vero E6 cells (Fig 3E). Despite reduction in the Nrf2 protein levels, the transcript of gene Nrf2 (*nfe2l2*) was not affected in our RNA-seq data (S3A Fig), which is consistent with the post-translational regulation of Nrf2 by Keap1 and proteasomal system [37]. Imbalanced redox metabolism was further supported by significant down-regulation of genes belonging to oxidoreductases, such as nitric oxide synthase-3 (*nos3*), glucose-6-phosphate dehydrogenase (*g6pd*), beta-carotene oxygenase 1 (*bco1*), and methyl sterol monooxygenase 1 (*msmo1*). As with our RT-qPCR data, infection with SARS-CoV-2 reduces the expression of genes associated with sulfur metabolism, which include $H_2S$ biogenesis (Fig 3D).

During SARS-CoV-2-HK infection, the most down-regulated cellular processes identified are carbon catabolism (TCA cycle, pyruvate metabolism, and fatty acid metabolism), OXPHOS (respiratory complex I to complex V), amino acid metabolism, and DNA replication/cell cycle (Fig 3D.). Since the metabolites of these processes are used by viruses for replication, our data agree with studies showing bioenergetic deficiency as one of the critical cellular responses to SARS-CoV-2 infection [33,38]. In this context, restoration of cholesterol and valine levels in infected cells results in elevated levels of virus, reinforcing the contribution of central metabolism to SARS-CoV-2 replication [39].

Next, we analysed transcriptional changes associated with GYY4137-mediated inhibition of SARS-CoV-2 replication. Treatment with GYY4137 led to altered expression of ~ 800 genes (fold change [FC] >1.5; false discovery rate [FDR] <0.1) compared to untreated VeroE6 cells (Fig 3F). We noticed that treatment with GYY4137 specifically reversed the effect of SARS-CoV-2-HK infection on the expression of genes associated with the oxidoreductase activity (e.g., *mthfr, acads, sc5d, hsd17b7, cbr4, msmo1, and me1)* and Nrf2- antioxidant pathway (Figs 3I and S3B). Furthermore, pathways related to defence and cellular response to viruses, including TLR, TNFα, Nod-like receptor, MAPK, RIG-1, and cytokine/chemokine signalling, were induced more in GYY4137-treated VeroE6 cells relative to untreated cells (Fig 3G and 3H).

We did not observe an effect of GYY4137 on the expression of central metabolism, respiration, and amino acid metabolism genes. We validated the expression changes of several Nrf2-specific antioxidant genes in response to GYY4137 by RT-qPCR (Fig 3J). The RT-qPCR and RNA-seq results were in agreement and indicated that antioxidant genes were down-regulated by SARS-CoV-2 infection, whereas GYY4137 treatment reversed this effect (Fig 3I and 3J). Altogether, H$_2$S supplementation induces a major realignment of redox metabolism and immune pathways associated with SARS-CoV-2 infection.

**SARS-CoV-2-mediated deregulation of redox metabolites was reversed by GYY4137**

To understand whether the expression changes induced by GYY4137 correlate with altered carbon and redox metabolites, we performed an untargeted metabolomics analysis on SARS-CoV-2-HK-infected VeroE6 cells for 24 h with and without treatment with 10 µM GYY4137 (Fig 4A). The metabolite profiles were analysed by multivariate principal component analysis (PCA). The metabolite profile separated SARS-CoV-2 infected from uninfected controls (SARS-CoV-2 vs UI) and GYY4137-treated (SARS-CoV-2 vs GYY4137) groups (Fig 4B). The analysis of the dataset indicated that central carbon metabolism (glycolysis, TCA cycle, and gluconeogenesis), amino acid metabolism (glycine, serine, aspartate, arginine, glutamate, proline, cysteine, and methionine), redox metabolism (glutathione, NAD+/NADH, and trans-sulfuration pathway metabolites), urea-ammonia cycle, and purine-pyrimidine metabolism were significantly enriched metabolic classes (*p* value < 0.1) (Fig 4C)

In SARS-CoV-2-HK infected Vero E6 cells, 70 metabolites were significantly altered (fold change [FC] >1.5; *p* value < 0.1). Consistent with virus entry and replication requiring nucleotide biosynthesis [40,41], we found a major enrichment of pyrimidine and purine metabolites (AMP, CDP, UDP and thymidine) (Fig 4D) in the infected VeroE6 cells. These data agree with a previous report on activation of de novo pyrimidine biosynthesis by the Nsp9 protein of SARS-CoV-2 as a mechanism to suppress the NF-κB dependent inflammatory response [40,42,43]. Glutaminolysis is another important source of carbon and nitrogen for satisfying the biosynthetic requirements of virus proliferation [44,45]. We detected an ∼2-fold accumulation of glutamate and depletion of glutamine upon SARS-CoV-2-HK infection. That suggested increased glutaminolysis to reload the TCA cycle by generating alpha-ketoglutarate (αKG) and citrate for sustained fatty acid and amino acid synthesis (Fig 4D). Increased cataplerosis of citrate derived from glutaminolysis likely re-routes carbon flux away from the TCA cycle to support fatty acid and amino acid biogenesis [44]. This finding is consistent with a marginal increase in citrate levels in virus-infected cells (Fig 4D). Our transcriptomics data revealed uniform suppression of the TCA cycle and mitochondrial respiration upon infection, suggesting a lack of ATP synthesis by OXPHOS. Interestingly, we found that, along with depletion of glutamine, cells accumulate succinate upon SARS-CoV-2-HK infection (Fig 4D). These findings suggest that down-regulation of OXPHOS forces cells to rely on glutamine to carry out substrate-level phosphorylation that generates αKG and succinate to sustain ATP production during infection [46]. An increased requirement to synthesize lipids, amino acids, and nucleotides is also reflected in a significant accumulation of serine upon infection (Fig 4D).

The association between SARS-CoV-2 infection and amino acid metabolism is reinforced by increased levels of tryptophan (TRP)-derived metabolites, such as kynurenine (KYN) and 3-hydroxykyneurine (Fig 4E), in the infected cells. The KYN metabolic pathway promotes the production of nicotinamide adenine dinucleotide (NAD+) from nicotinamide through a salvage pathway [47–50]. We detected an ∼4-fold increase of NAD+ precursor, nicotinamide, and an ∼2-fold increase in NADH levels (Fig 4E) in SARS-CoV-2-HK-infected cells compared to the uninfected condition. The data agree with earlier reports that SARS-CoV-2 infection deprives the host of NAD+ by diminishing its principle biosynthetic pathway from quinolinic acid, while promoting NAD+ acquisition through a salvage pathway. This deregulation of TRP and NAD metabolites likely contribute to inflammation, apoptosis, and disease severity in COVID-19 patients [49]. These findings agree with the studies showing higher KYN:TRP ratio in COVID-19 patients [51–53].

In agreement with our RNA-seq data showing down-regulation of GSH biogenesis and sulfur metabolism, levels of methionine, cysteinyl-glycine, and glutamyl-cysteine were significantly depleted upon SARS-CoV-2 infection

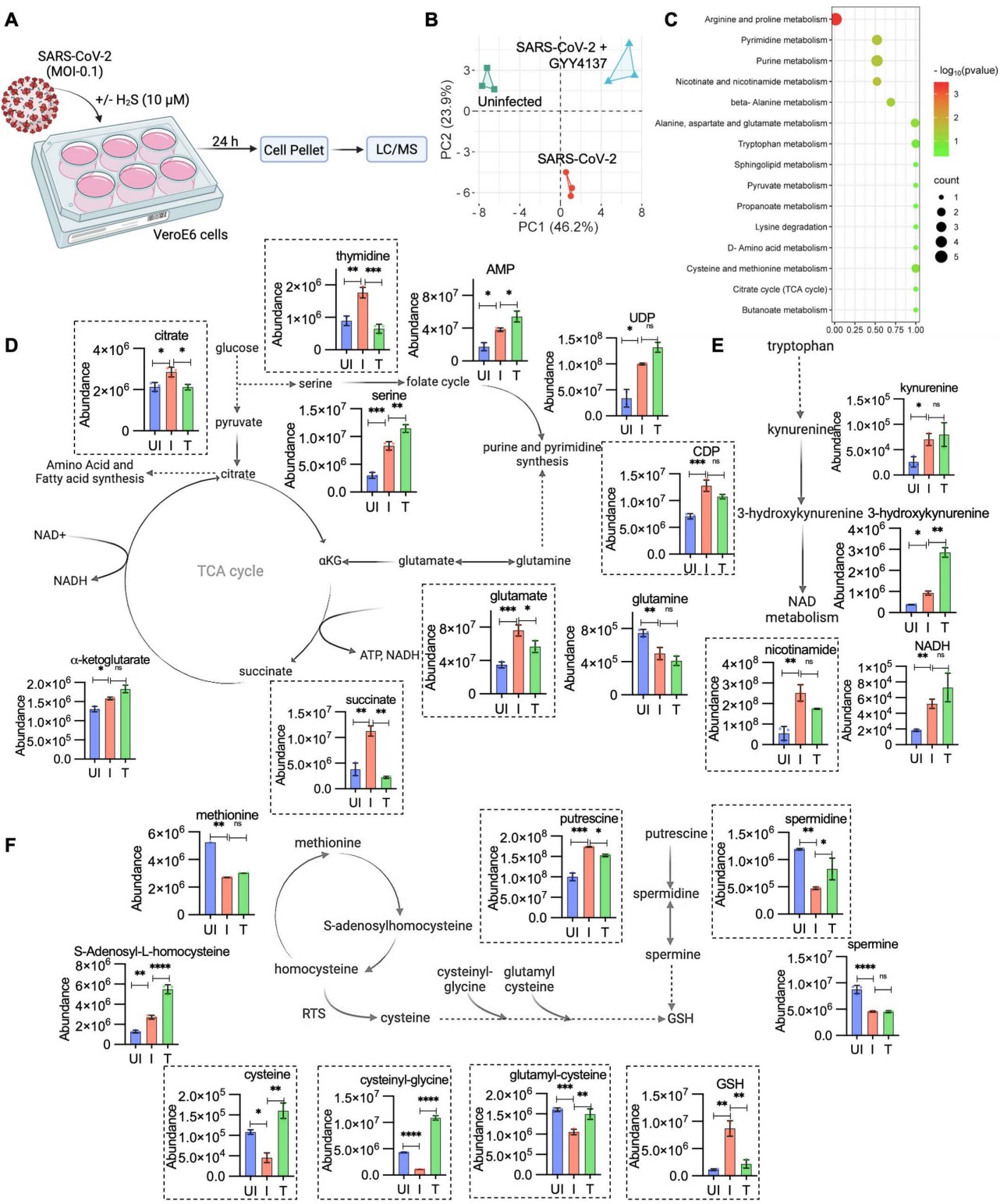

**Fig 4. SARS-CoV-2 mediated deregulation of redox metabolites are reversed by GYY4137:** (A) Schematic of LC/MS based analysis of cellular metabolites. (B) PCA analysis plot of metabolites in uninfected and SARS-CoV-2-HK infection in presence or absence of 10 μM GYY4137. (C) Pathway enrichment analysis by Metaboanalyst 6.0 on differently expressed metabolites. (D,E,F) Fold change of intracellular TCA cycle intermediates, nucleotide

metabolism, one carbon metabolism, tryptophan metabolism, reverse transulfuration pathway. Box highlights reversely expressed metabolite with GYY4137 treatment (UI- Uninfected, I- Infected, T- Infected and treated with GYY4137). Results are expressed as mean ± standard deviation of data from single experiment, performed in three technical replicates. *$p < 0.05$; ** $p < 0.01$; *** $p < 0.001$; **** $p < 0.0001$, by student's t-test or one-way ANOVA with multiple comparison using Tukey's method. Fig 4A graphic created in Biorender, https://BioRender.com/ttos191.

(Fig 4F). Since these amino acids are precursors for biogenesis of $H_2S$ and GSH [18,54,55], our results are consistent with decreased $H_2S$ production upon SARS-CoV-2 induction. Surprisingly, although the GSH precursors (cysteinyl-glycine and glutamyl-cysteine) were down-regulated, the total GSH levels increased by 3.5-fold in the infected cells. It is likely that while GSH biogenesis enzymes are down-regulated, cells still accumulate GSH by decreasing GSH catabolism/degradation as a way to counteract virus-induced oxidative stress. SARS-CoV-2 infection also resulted in accumulation of serine, which lies at the central branch point linking biosynthetic flux from glycolysis to GSH and the one-carbon metabolic cycle (Fig 4D) [56,57]. Accumulation of serine and active methyl cycle intermediate S-adenosylhomocysteine (SAH) indicate defects in redox balance and methylation potential of infected cells [58–61]. Furthermore, we observed a significant effect of virus infection on polyamines, such as putrescine (1.6-fold upregulation) and spermine/spermidine (2-fold down-regulation). Because polyamines are known scavengers of ROS and stimulate GSH production [62–65], their deregulation correlates well with an overall change in redox metabolites upon virus infection.

We next assessed the effect of GYY4137 on metabolic changes associated with SARS-CoV-2-HK infection. Upon GYY4137 treatment, approximately 76 metabolites were significantly altered (fold change [FC] >1.5; $p$ value < 0.1). More-importantly, GYY4137 treatment reversed the influence of virus infection on a specific set of 28 metabolites related to redox homeostasis. For example, precursors for GSH biosynthesis (cysteinyl-glycine, glutamyl-cysteine, and cysteine) were upregulated or restored to normal levels (Fig 4F). Similarly, the levels of spermidine were elevated by GYY4137. The increase in pyrimidines (thymidine), amino acids (glutamate), citrate, and succinate upon virus infection was significantly reversed by GYY4137 treatment. Overall, supplementation with this $H_2S$ donor restored GSH homeostasis and normalized deregulated nucleotides and amino acid pools.

## GYY4137 prevents redox imbalance and restores mitochondrial function upon SARS-CoV-2 infection

Our gene expression data indicate activation of the Nrf2 pathway, which protects from oxidative stress, as a major cellular response of virus infected cells to GYY4137. Therefore, we asked whether GYY4137 suppresses SARS-CoV-2-induced oxidative stress. First, we quantified oxidative stress during SARS-CoV-2 infection in the cytosol and mitochondria of VeroE6 cells. To do this, we created stably transfected VeroE6 cells that express a genetically encoded redox biosensor (Grx1-roGFP2) in the cytoplasm (Vero-Cyto-Grx1-roGFP2) or mitochondria (Vero-Mito-Grx1-roGFP2) (Fig 5A and 5B) [66–69]. The Grx1-roGFP2 biosensor allows detection of dynamic changes in the cellular and subcellular redox environment by quantifying the redox potential of the major cellular thiol glutathione (GSH/GSSG) [67]. Grx1-roGFP2 has two fluorescence excitation maxima at 405 and 488 nm with a common emission at 510 nm [67]. An increase in oxidative stress increases the ratio of 405/488, whereas the biosensor exhibits an inverse response upon reduction [67]. We confirmed the accurate subcellular localization of Grx1-roGFP2 in either cytosol or mitochondria in VeroE6 cells (Fig 5B). We then infected these cells with SARS-CoV-2-HK and monitored ratiometric changes in 405/488 by flow cytometry. A time-dependent 405/488 ratio increase was observed in cyto-Grx1-roGFP2 and mito-Grx1-roGFP2 upon infection (Fig 5C), indicating an increase in oxidative stress. Mitochondria displayed a higher biosensor response upon infection, indicating that the virus induces significantly more oxidative stress in mitochondria than in the cytosol. As expected, pretreatment with GYY4137 completely abrogated oxidative stress induced by SARS-CoV-2 in the cytosol and mitochondria of VeroE6 cells (Fig 5D).

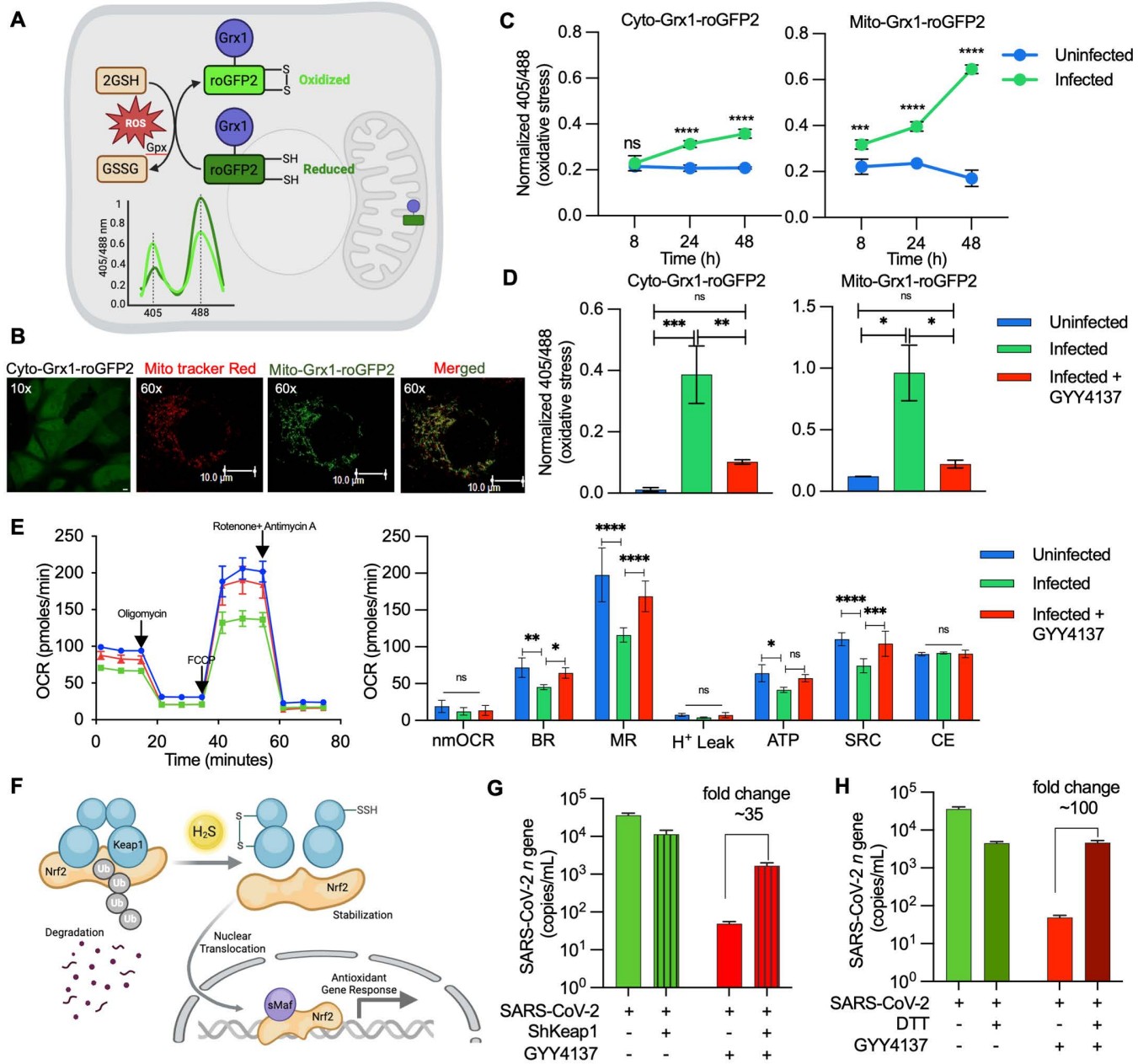

**Fig 5. GYY4137 prevents redox imbalance and restores mitochondrial function upon SARS-CoV-2 infection:** (A) Schematic representation of Grx1-roGFP2 oxidation and reduction in response to ROS inside a mammalian cell stably expressing the biosensor. (B) Confocal images of VeroE6 cells stably expressing Grx1-roGFP2 in cytoplasm and mitochondria. Grx1-roGFP2 is shown as green, mitotracker stain is red, and yellow signal demonstrates overlap. Scale bar represents 10 μm. (C) Biosensor-expressing cell lines were infected with 0.01 MOI of SARS-CoV-2-HK. Biosensor ratio (405/488 nm) was measured by flow cytometry at indicated time points. (D) Biosensor ratio (405/488 nm) of SARS-CoV-2-HK infected cells in presence or absence of 5 mM GYY4137 at 48 h pi. (E) OCR measurement of VeroE6 cells infected with 0.1 MOI SARS-CoV-2-HK in presence or absence of 10 μM GYY4137. Various respiratory parameters were derived from OCR values. nmOCR- non-mitochondrial oxygen consumption rate, BR- basal respiration, MR- maximal respiration, H+ Leak – proton leak, SRC-spare respiratory capacity, and CE- coupling efficiency. (F) Schematic representation of Nrf2 regulation by Keap-1 inside mammalian cells (Ub- Ubiquitin, sMaf- small musculoaponeurotic fibrosarcoma). (G) VeroE6 and Vero-shKeap1 cells were infected with 0.01 MOI SARS-CoV-2-HK in presence or absence of 5 mM GYY4137; virus replication was measured by RT-qPCR at 24 h pi. (H) VeroE6 cells treated with 10 μM DTT along with 5 mM GYY4137, prior to infection with 0.01 MOI SARS-CoV-2-HK. Virus replication was measured by RT-qPCR at 24 h pi. Results are expressed as mean ± standard deviation; representative of data from two independent experiments, performed thrice. * $p < 0.05$; ** $p < 0.01$; *** $p < 0.001$; **** $p < 0.0001$, by two-way ANOVA with multiple comparison using Tukey's method (C,E,G,H) or multiple student's t-test (D). (See also S4 Fig). Fig graphics created in Biorender, https://BioRender.com/iovctrx (Fig 5A), https://BioRender.com/wgzverk (Fig 5F).

Our findings agree with reports supporting the role of $H_2S$ in redox balance, GSH homeostasis, and mitochondrial function [2,70,71]. $H_2S$ could restore mitochondrial bioenergetics by acting as a substrate for the electron transport chain [12,72]. Since RNA-seq data during infection indicate diminished expression of OXPHOS genes, and the bio-sensor data showed heightened mitochondrial stress, we reasoned that GYY4137 could maintain redox balance by counteracting mitochondrial dysfunction induced by SARS-CoV-2. We studied the effect of GYY4137 on mitochondrial function using a Seahorse XF Extracellular Flux Analyzer (Agilent) as described previously by us [69,70]. Both basal and ATP-coupled respiration was significantly lower in VeroE6 infected with SARS-CoV-2-HK (Fig 5E). Upon dissipation of the mitochondrial proton gradient by FCCP, the maximal respiration capacity was markedly diminished, resulting in the exhaustion of spare respiratory capacity (SRC). Consistent with the role of $H_2S$ in reducing cytochrome-c oxidase for respiration [73], GYY4137 treatment reversed the respiratory indicator of infected cells to nearly uninfected levels (Fig 5E). These results indicate that SARS-CoV-2 infection decelerates respiration and diminishes the capacity of VeroE6 cells to respirate maximally. Significantly, treatment with GYY4137 restored all essential parameters reflecting mitochondrial health (Fig 5E).

The above findings and our expression data indicate that GYY4137 mobilizes cellular antioxidant machinery, such as the Nrf2 pathway, in response to virus infection. $H_2S$ has been consistently shown to activate Nrf2 via S-persulfidation of its repressor, Kelch-like ECH-associated protein 1 (Keap1) [74,75]. Under normal growing conditions, Keap1 binds to Nrf2 and promotes Nrf2 degradation through proteasomal machinery. In response to $H_2S$ exposure, Nrf2 dissociates from the S-persulfidated form of Keap1, translocates to the nucleus, and induces the expression of antioxidant genes [76] (Fig 5F). Nrf2 activation also reduces mitochondrial ROS by promoting mitochondrial respiration [77]. Based on these ideas, we determined whether the underlying mechanism of $H_2S$-mediated suppression of SARS-CoV-2 replication involves the Keap1-Nrf2 axis. Since $H_2S$ sensing is mediated through S-persulfidation of Keap1 to activate Nrf2, we expected that reducing the expression of Keap1 would interfere with the activation of Nrf2 by GYY4137. We depleted endogenous Keap1 levels in VeroE6 cells using RNA interference (Vero-shKeap1) and confirmed the de-repression (~2 fold) of Nrf2-dependent antioxidant genes (ho-1 and txnrd1) in the knockdown strain (S4A Fig) [78]. Moreover, while GYY4137 treatment reduces virus replication by ~ 700-fold in VeroE6 cells, a similar treatment resulted in only ~7-fold reduction in SARS-CoV-2-HK levels in Vero-shKeap1. That resulted in an approximately 35-fold activity difference for GYY4137 between VeroE6 and Vero-shKeap1 cells (Fig 5G).

In addition to S-persulfidation, $H_2S$ could promote disulfide bond formation between the two Keap-1 molecules [74,79]. Therefore, as an additional verification, we treated VeroE6 cells with a cell-permeable inhibitor of disulfide formation, 1,4-dithiothreitol (DTT), and showed that the virus-suppressing effect of GYY4137 is reduced (Fig 5H). These findings are consistent with our RNA-seq data and indicate that $H_2S$ likely prevents virus replication by activating the Nrf2-specific redox response via Keap1 persulfidation or disulfide bond formation.

## GYY4137 controls SARS-CoV-2 replication in animal models of infection

Having shown that $H_2S$ suppresses SARS-CoV-2 replication in multiple cell lines, we next examined the ability of GYY4137 to limit virus proliferation in mice and hamsters. Balb/c mice (n = 10) were treated with 50 mg/kg GYY4137 or vehicle control (PBS) 1 h before infection and 6 and 24 h post-infection with MA-10 (mouse-adapted SARS-CoV-2) [80,81]. Mice were intranasally infected with $5 \times 10^4$ PFU of SARS-CoV-2 (Fig 6A). Virus replication was determined in lung tissues by RT-qPCR and plaque assay at days 3 and 5 p.i.. At all-time points, administration of GYY4137 uniformly reduced (~100 fold) viral load as compared to the vehicle control (Fig 6B). Histopathological changes in the lungs were proportionate to differences in the viral burden. Virus-infected control animals showed increased cellular infiltration in perivascular and peri-bronchial spaces with significant areas of consolidation and necrosis (pathology score = 3–4). In contrast, the extent of pulmonary damage was negligible in GYY4137-treated animals (pathology score = 1) (Fig 6C and 6D). We also confirmed that GYY4137 treatment induced the expression of antioxidant (Nrf2-regulon) genes and reduced the expression of inflammatory genes in the lungs

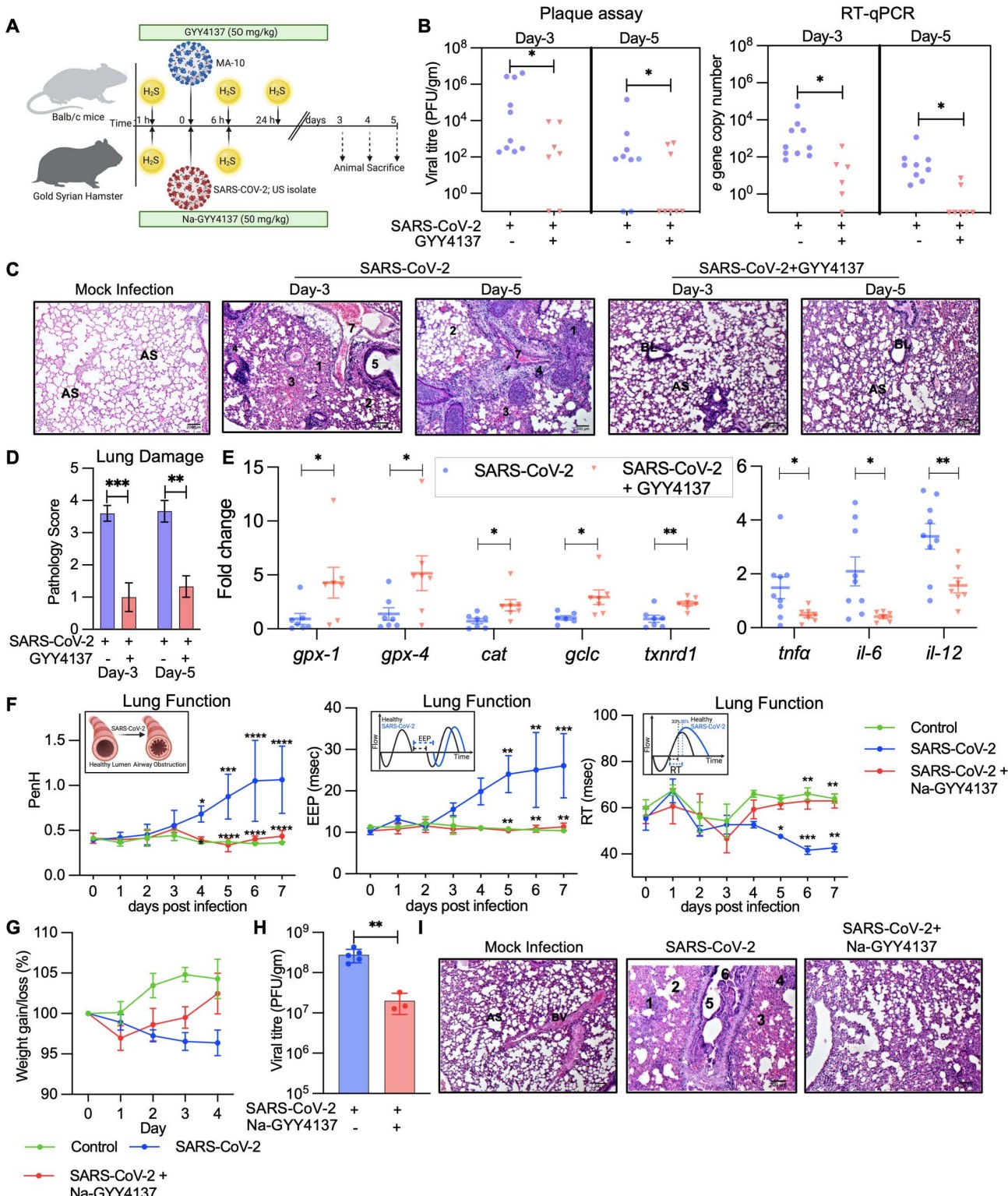

**Fig 6. GYY4137 controls SARS-CoV-2 replication in animal models of infection:** (A) Schematic representation of animal experiments. (B) Balb/c female mice were treated with 50 mg/kg body weight GYY4137 or vehicle and infected with mouse-adapted SARS-CoV-2 (MA-10). Mice were sacrificed at day -3 and day -5 to determine viral load by plaque assay, expressed as pfu per gram of lung tissue and by RT-qPCR. (C,D) Hematoxylin- and

eosin-stained lung sections from different mice groups, with pathology score. (E) Lung tissues were processed to isolate total RNA. Anti-oxidant and anti-inflammatory gene response was measured by RT-qPCR. (F) Balb/c female mice were treated with 50 mg/kg body weight Na-GYY4137 or vehicle and infected with mouse-adapted SARS-CoV-2 (MA-10). Lung function was measured daily by unrestrained whole body plethysmography. Inset figure explains respective lung function parameter. (G,H) Gold Syrian hamsters were treated with 50 mg/kg body weight Na-GYY4137 or vehicle and infected with SARS-CoV-2 (US variant). Body weight and other clinical parameters were measured daily (G). Animals were sacrificed at day 4 to determine viral load by plaque assay (H). (I) Hematoxylin- and eosin-stained lung sections from different hamster groups. (Annotations for C and I) (1) acute alveolar inflammation and consolidation of alveolar space (pneumonia) with leucocytic alveolitis; (2) diffuse, damaged, and distended alveolar space; (3) severe blood hemorrhage; (4) necrosis; (5) acute bronchiolitis; (6) breakage of bronchial epithelial linings; (7) perivascular inflammation and vascular congestion. (BL) bronchial lumen (AS) alveolar space (BV) blood vessel. * $p < 0.05$; ** $p < 0.01$; *** $p < 0.001$ by student's t-test with Welch's correction or two-way ANOVA with multiple comparison using Tukey's method (F). (See also S5 Fig). Fig graphics created in Biorender, https://BioRender.com/1by4ezc (Fig 6A), https://BioRender.com/zhpdqda (Fig 6F).

of infected mice (Fig 6E). We also examined the effect of Na-GYY4137 in mice infected with MA-10. Similar to GYY4137, we found that Na-GYY4137 uniformly reduced the viral load in MA-10 infected mice (S5A Fig).

Interestingly, mouse-adapted SARS-CoV-2 (MA-10) recapitulates multiple aspects of severe COVID-19 disease, including altered pulmonary function linked to acute lung injury (ALI) and acute respiratory distress syndrome (ARDS) [80]. Therefore, we measured the impact of infection and GYY4137 treatment on lung physiology using unrestrained whole-body plethysmography (uWBP) that non-invasively quantifies pulmonary function in spontaneously moving mice. As compared to control mice, infected mice experienced a loss in pulmonary function as indicated by a significantly elevated pause (PenH), a measurement of airway obstruction (Fig 6F) [82]. The end-expiratory pause (EEP), which provides a measurement of the time between the end of exhalation and the active start of inspiration [83], was notably prolonged between 5–7 days p.i (Fig 6F). Infected animals also show decreased relaxation time (RT), a measurement of the time required to exhale a fixed percentage (30%) of tidal volume (Fig 6F). Other pulmonary functions, such as peak expiratory flow (PEF) and mid-tidal expiratory flow (EF50), also increased upon infection, although this trend did not reach significance (S5B and S5C Fig). More importantly, Na-GYY4137 significantly mitigated the loss of pulmonary function (PenH, EEP, and RT) observed in the infected animals (Fig 6F).

We also assessed GYY4137 efficacy in the gold Syrian hamster model. In this infection model, we explored the potential of Na-GYY4137 to be an $H_2S$ donor owing to its lower toxicity [23]. Similar to mice, we treated animals with 50 mg/kg of Na-GYY4137 or vehicle control 1 h before infection and 6 h post-infection with $10^5$ PFU SARS-CoV-2 (hCoV-19/USA/MD-HP05285/2021; US isolate). As reported earlier, a gradual reduction in animal weight was observed over time in virus-infected animals [84] (Fig 6G). In contrast, following an initial decline at day 1 post-infection, Na-GYY4137-treated animals showed a gradual recovery in weight at days 2, 3, and 4 post-treatment. In line with this, administration of Na-GYY4137 led to a 10-fold lower viral load as compared to the vehicle control (Fig 6H). The magnitude of lung damage was highest in SARS-CoV-2-infected, vehicle-treated animals (pathology score = 3–4), intermediate in Na-GYY4137-treated animals (pathology score = 1), and negligible in mock-infected animals (pathology score = 0) (Fig 6I). In sum, exposure to GYY4137 results in potent inhibition of viral replication and improved disease pathology *in-vivo*.

## Discussion

The main conclusion of our study is that SARS-CoV-2 multiplication correlates with the depletion of endogenous $H_2S$, which is associated with the breakdown of redox homeostasis and mitochondrial bioenergetics, in particular, diminished OXPHOS, energy metabolism, and mitochondrial oxidative stress. Decreased $H_2S$ resulted in the altered expression of genes involved in maintaining Nrf2-dependent cellular redox potential and the innate immune response. Several cell-line models of SARS-CoV-2 replication, nasopharyngeal samples from COVID-19 patients, and two animal infection models support our conclusions. Pharmacological complementation with GYY4137 confirmed the role of $H_2S$ in effectively suppressing SARS-CoV-2 by restoring redox balance, carbon metabolism, amino acid metabolism, and mitochondrial function. Lastly, restoring $H_2S$ levels inhibited virus replication and restored pulmonary function in mice and hamsters.

Collectively, the data show that $H_2S$ deficiency promotes virus replication and that the $H_2S$ donors GYY4137 and Na-GYY4137 counteract virus proliferation. Our findings highlight that $H_2S$-based interventions can be exploited to maintain cellular homeostasis and reduce SARS-CoV-2 spread during infection.

How does $H_2S$ promote suppression of SARS-CoV-2 proliferation? Several studies reveal that SARS-CoV2 replication is associated with mitochondrial dysfunction, redox imbalance, inflammation, and NF-κB/TLRs/TNF signalling pathways [13,34,85–87]. $H_2S$ corrects several of these biological dysfunctions. For example, $H_2S$ in physiological concentrations improves mitochondrial health, protects from oxidative stress, and reverses inflammation [18,11,72,88]. Our transcriptomics, metabolomics, and XF flux measurements support a mechanism whereby virus infection diminishes endogenous $H_2S$, resulting in loss of mitochondrial function and oxidative stress. Consistent with this, genetic depletion of endogenous $H_2S$ promotes virus proliferation, suggesting that maintaining $H_2S$ levels controls SARS-CoV-2 infection. As expected, supplementing $H_2S$ with GYY4137 encourages the maintenance of mitochondrial health and redox homeostasis as a control of SARS-CoV replication. $H_2S$ is known to modulate mitochondrial activity and redox balance at low concentrations through multiple mechanisms including via the Nrf2/Keap1 axis [37,89]. Our transcriptomic data point toward the role of Nrf2/Keap1 as an upstream mediator of $H_2S$ signalling in SARS-CoV-2 infection. $H_2S$ is shown to directly induce S-persulfidation of Keap1, which promotes Nrf2 nuclear translocation and stimulation of antioxidant genes expression. The role of Nrf2-Keap1 was further strengthened by our demonstration that depletion of Keap1 significantly abrogated the virus suppressive effect of GYY4137. Treatment of cells with DTT that reverses S-persulfidation also decreases the effect of GYY4137 on virus inhibition. Our data also support a primary role of the Keap1-Nrf2 pathway in mediating the antiviral effect of GYY4137.

Several other viruses, e.g., human immunodeficiency virus (HIV), respiratory syncytial virus (RSV) are known to modulate the expression of Nrf2 thereby disrupting redox homeostasis; restoration of Nrf2 function suppresses virus proliferation [70,90]. In the context of SARS-CoV-2, mice lacking the nuclear factor erythroid 2-related factor 2 *(nrf2)* gene exhibit severe disease, increased lung inflammation, and elevated virus titer [91]. Our study showed that $H_2S$-mediated restoration of Nrf2 activity decreases SARS-CoV-2 replication, indicating that Nrf2 is protective during this viral infection.

$H_2S$ also exerts its effect by reacting with disulfides and sulfenic acid to form highly nucleophilic persulfides (RSSH)[92]. We suspect that formation of persulfides on viral and host proteins upon exposure to $H_2S$ donors could directly regulate SARS-CoV-2 infection. The activity of several viral proteins, such as spike glycoprotein (S protein) and thiol proteases (PLpro and 3CLpro), depend on disulfide bond formation [93–97], which $H_2S$ can convert to persulfides. Consistent with this idea, SARS-CoV-2 PLpro and 3CLpro were inactivated via persulfidation induced by glutathione trisulfide (GSSSG) [27]. Likewise, disulfide bond cleavage at the receptor-binding domain (RBD) of the S protein likely suppresses SARS-CoV-2 infection [98]. All of this could influence virus entry, which is consistent with our findings showing that pre-treatment with GYY4137 had a significant impact on SARS-CoV-2 replication. NO may have a similar anti-SARS-CoV-2 effect by inactivating viral proteases, possibly through S-nitrosylation [99]. Interestingly, a recent study found a synergistic effect of NO and S-persulfidation in inhibiting SARS-CoV-2 PLpro [27].

Previous clinical studies have proposed an inverse relationship between endogenous $H_2S$ levels and the severity of COVID-19 [100], indicating the beneficial potential of $H_2S$-releasing drugs for COVID-19 treatment [101]. The therapeutic potential of $H_2S$ on the COVID-19 pathogenesis was largely inferred by the pharmacological introduction of sulfur-containing compounds such as diarlytrisulfide and S8 or reactive hydroperssulfides, which include cysteine persulphide (CysSSH), glutathione persulphide (GSSH) and oxidized glutathione trisulphide (GSSSG) [27]. However, prior to the present work, there had been no demonstration of the direct contribution of endogenous $H_2S$ and the impact of a slow-releasing $H_2S$ donor for anti-SARS-CoV-2 effects, particularly the therapeutic effects in mice and hamster infection models. In this context, it is crucial to investigate the clinically-relevant and safe concentrations of the $H_2S$ donors. Exposure to 5 mM of GYY4137 resulted in ~600 µM of $H_2S$ inside cells. At this concentration, we did not observe any cytotoxicity but found excellent inhibitory effect on virus proliferation and a larger impact on transcriptome. Low micromolar concentration of GYY4137 exhibited uniform influence on mitochondrial bioenergetics and steady-state metabolites. Future metabolite flux-based experimentations are

needed to examine the impact of varying H$_2$S concentration on virus suppression, and to correlate transcriptomic data with the metabolomic changes. The absorption and metabolism of H$_2$S ensures sulfide homeostasis upon administration of high H$_2$S doses. The sulfide catabolism pathways promote oxidation of excess H$_2$S to thiosulfate and sulfate, which are cleared through excretory system in free or conjugated form [102]. S-adenosyl methionine (SAM)-dependent methylation of H$_2$S by human methyltransferase-like protein 7B (METTL7B) could be another mechanism to reduce potential toxicity [103]. All of these observations would have led to successful application of high doses of GYY4137/Na-GYY4137 in suppressing virus proliferation and lung pathology in mice and hamsters.

Several modulatory effects of H$_2$S on immune responses have been reported [104,105]. Mechanistically, multiple signal transduction pathways serve as a molecular target for H$_2$S. These include NF-κB, AKT, AMP Kinase, type I/II interferons, JAK-STAT, and PTEN for immune regulation [104,106]. In our RNA-seq data, many signalling components (NF-κB, type I/II interferon, JAK-STAT) were induced in response to SARS-CoV-2 infection and were induced even more by GYY4137 treatment. Using HIV infection as a model, we previously demonstrated that the H$_2$S donor GYY4137 modulates the expression of genes associated with oxidative stress, inflammation, antiviral response, and apoptosis. Many of these effects were likely mediated by H$_2$S-mediated suppression of p65 ser-536 phosphorylation, a major NF-κB subunit, and Nrf2 activation [70]. These data are consistent with our findings in animal lungs where SARS-CoV-2 infection showed up-regulation of proinflammatory cytokines and down-regulation of Nrf2-dependent antioxidant genes, with GYY4137 treatment reversing these effects. H$_2$S may, therefore, control inflammatory over-reactions occurring during SARS-CoV-2 infection and maintain antioxidant balance to provide broad protection against viral and inflammatory diseases, such as COVID-19.

Our recent work with HIV also demonstrates that endogenous H$_2$S biogenesis by CTH and 3-MST distinctly affects virus proliferation. While CTH activity was required to suppress HIV, activation of 3-MST resulted in virus proliferation [70,71]. The current study showed that SARS-CoV-2 infection efficiently down-regulates the production of H$_2$S by reducing expression of CBS, CTH, and 3-MST enzymes. However, the suppression of only CTH promoted virus proliferation, indicating a major role for CTH-mediated H$_2$S production in controlling SARS-CoV-2 replication in VeroE6 cells. Future experiments are needed to understand the role of CTH in controlling SARS-CoV-2 in diverse cells, animal models, and humans. Moreover, studies have shown that CBS/CTH/3-MST triple knock-out mice do not show appreciable changes in several sulfur metabolites [107]. Therefore, while CBS/CTH/3-MST might be responsible for generating H$_2$S and persulfides, additional pathways such as CARS2/CPERS could be necessary for producing endogenous persulfides *in-vivo* [108].

In conclusion, we identified H$_2$S as a central factor in SARS-CoV-2 infection. Our systematic mechanistic dissection of the role of H$_2$S in cellular bioenergetics, redox metabolism, and virus replication unifies many previous phenomena associated with various viral infections, including COVID-19, and with chronic lung diseases such as chronic obstructive pulmonary disease (COPD).

## Materials and methods

### Ethics statement

Clearances for this project have been obtained from the Institutional Biosafety Committee (IBSC; Approval number: IBSC/IISc/AS/16/2020), Institutional Human Ethics Committee (IHEC; Approval number: 13–11092020) and Institutional Animal Ethics Committee (IAEC; Approval number: CAF/ETHICS/940/2023). For use of human samples, informed verbal consent was obtained from each participant, before the study.

### Cell lines and virus

HEK293T, VeroE6 cell, Calu-3 cells and HEK293T cells expressing human ACE2 cells were cultured in complete media containing Dulbecco's modified Eagle medium (Cell Clone, Genetix, India) with 10% CELLECT FBS (MP Biomedicals), 100 IU/mL penicillin, 100 µg/mL streptomycin and 0.25µg/mL amphotericin-B (Sigma-Aldrich). Calu-3 cells were additionally supplemented with non-essential amino acids (Sigma-Aldrich). All cell lines were maintained at 37°C in a humidified

incubator with 5% $CO_2$. All cell lines are verified to be mycoplasma free using a EZdetect PCR detection kit for Mycoplasma (HIMEDIA).

SARS-CoV-2 isolates (Hong Kong/VM20001061/2020, NR-52282; hCoV-19/USA/MD-HP05285/2021, NR-55671 (Delta Variant); Mouse-Adapted MA-10 variant, NR-55329) were obtained from BEI Resources, NIAID, NIH and were propagated and titered using standard plaque assay in VeroE6 cells [109]. All experiments with live SARS-CoV-2 virus were performed in Viral Biosafety level-3 laboratory at CIDR, IISc.

## Animals

Balb/c mice and Gold Syrian hamsters were obtained from CPCSEA (The Committee for the Purpose of Control and Supervision of Experiments on Animals)-approved animal dealers (IAEC approval number- CAF/ETHICS/940/2023; CPCSEA registration number of dealer- 2076/PO/RcBiBt/S/19/CPCSEA). Animals were tested to be free from any pathogens. All the animals were acclimatized to laboratory conditions before using them for experiments. Animals were disposed by following CPCSEA guidelines.

## Chemical reagents

Sodium hydrosulphide (NaHS), morpholin-4-ium 4-methoxphenyl(morpholino) phosphinodithioate dichloromethane complex (GYY4137), sodium-GYY4137 (Na-GYY4137), D,L-propargylglycine (PAG), dithiothreitol (DTT), L-cysteine and thiazolyl blue tetrazolium bromide (MTT) were purchased from Sigma-Aldrich. EZ-Link Iodoacetyl-PEG2-Biotin (IAB), Dynabeads-M280 Streptavidin and Lipofectamine 3000 were purchased from ThermoFisher Scientific.

## Generation of stable cell lines

ShRNA constructs against CTH, CBS, 3-MST and Keap1 were obtained from RNAi consortium (TRC) library (Sigma-Aldrich) (S1 Table). Grx1-roGFP2 with or without the mitochondrial target sequence (cox8a) was cloned into the pLVX lentiviral expression vector system. For generating stable cells, HEK293T cells were transfected with the target plasmids along with packing plasmids (psPax2 and pMD2 G) using lipofectamine 3000 as per the manufacturer's protocol. Forty-eight h post transfection, cell supernatant containing lentiviral particles was used for transducing VeroE6 cells, supplementing with 10 μg/mL polybrene. Stably transduced VeroE6 cells were selected in the presence of 3 μg/mL puromycin.

## Cytotoxicity assay

Cytotoxicity of the compounds was assessed by MTT-based assay, as described previously [110]. Briefly, 20,000 cells were seeded in a 96 well plate and incubated at 37°C, 5% $CO_2$ in a humidified incubator overnight. Next day, cells were treated with two- fold serially diluted compounds in duplicate wells for 48 h before addition of 0.8 mg/mL MTT substrate and further incubated until the appearance of MTT crystals (~1 h). Crystals were dissolved in DMSO, and absorbance was read at 595 nm with a reference filter of 620 nm using a SpectraMax M3 plate reader (Molecular Devices).

## Mitochondria staining

Grx1-roGFP2-expressing cells were fixed with 4% PFA. For staining mitochondria prior to fixing, the cells were pretreated for 1 h with 100 nM MitoTracker (Invitrogen). The coverslips were washed thoroughly with PBS and mounted onto glass slides with mounting media (Antifade reagent, Invitrogen). Grx1-roGFP2 fluorescence was analysed at 488 nm excitation and 525 nm emission, and Mito-Tracker-stained cells were visualized at 540 nm excitation and 630 nm emission.

## Infection of cell lines

Cells were seeded in 24- or 12-well plates such that the density at the time of infection was 90–95%. For all drug experiments, cells were either pretreated with 5 mM GYY4137 or 20 mM PAG or left untreated for 4 h (2 h for PAG). Viruses

at the indicated MOI were added to the cells (total volume 200 µL) and incubated for 1 h with intermittent mixing. Virus inoculum was removed after 1 h, and infection medium (DMEM with 2% FBS), restoring the initial dose of the compounds, was added to the cells which were incubated for the indicated times. After incubation, cell supernatant was processed for plaque assay and viral RNA isolation (mdiViral isolation kit). Cells were lysed and processed separately for western blotting and RNA isolation (Qiagen RNAeasy RNA isolation kit).

### Reverse transcription-quantitative polymerase chain reaction (RT-qPCR)

Total RNA was extracted using RNeasy mini kit (Qiagen). The synthesis of cDNA was performed with 500 ng of total RNA in a 20 µL reaction, using iScript Reverse Transcription Supermix (Bio-Rad). cDNA was subjected to quantitative real-time PCR (iQ SYBR Green Supermix, Bio-Rad), performed using the Bio-Rad C1000 real-time PCR system. 18S rRNA, actin, or beta-2-microglobulin was used as the housekeeping gene for normalization (S2 Table). The Ct values were analysed using the delta-delta Ct method ($2^{-\Delta\Delta Ct}$ method). SARS-CoV-2 viral copies were calculated by generating a standard curve against the viral N gene.

### Western Blot analysis

Total cell lysates were prepared using radioimmunoprecipitation (RIPA) lysis buffer (50 mM Tris [pH 8.0], 150 mM NaCl, 1% Triton X-100, 1% sodium deoxycholate, 0.1% SDS [sodium dodecyl sulfate], 1 × protease inhibitor cocktail (Sigma-Aldrich), and 1 × phosphatase inhibitor cocktail [Sigma-Aldrich]). After incubation on ice for 20 min, the lysates were centrifuged at 15,000 x g 4°C for 15 min. Clarified supernatant was taken, and total protein concentration was determined by Bicinchoninic Acid Assay (Pierce, ThermoFisher Scientific). Total protein extracts were separated by SDS-PAGE and transferred onto polyvinylidene difluoride (PVDF) membranes. After transfer, the membranes were blocked with 5% skim milk. Membranes were probed with anti-CBS (EPR8579), CTH (ab151769), MST (ab154514), from Abcam; anti-SARS-CoV-2 Nucleocapsid (940901) from Biolegend; anti-Nrf2 (CST-12721), and GAPDH (CST-97166) from Cell Signaling Technologies, Inc; and anti-rabbit IgG (CST-7074) and anti-mouse IgG (CST-7076) were used as secondary antibodies. Proteins were detected by ECL and visualized by chemiluminescence (PerkinElmer, Waltham, MA) using the Bio-Rad Chemidoc Imaging system. For membrane re-probing, stripping buffer was used (2% SDS [w/v], 62 mM Tris-Cl buffer [0.5 M, pH 6.7], and 100 mM β-mercaptoethanol) for 20 min at 55°C. After extensive washing with phosphate- buffered saline (PBS) containing 0.1% Tween 20 (Sigma-Aldrich), membrane was blocked and re-incubated with desired antibodies.

### Plaque assay

Plaque assay was done to measure the infectious virus counts as described previously [109]. Briefly, VeroE6 cells were seeded in a 6-well plate, such that the plates were 95% confluent at the time of infection. Cells were washed once with 1x PBS and inoculated with 200 µL of dilutions of cell culture supernatant (containing virus) and allowed to infect for 1 h with intermittent rocking. The virus inoculum was then removed, and the cell monolayer was overlaid with DMEM containing 2% FBS and 0.8% agarose (Sigma-Aldrich). After 72 h, cells were fixed with 4% paraformaldehyde (PFA), and plaques were visualized by crystal violet (Sigma-Aldrich) staining. The viral titre was calculated as follows: Titre of SARS-CoV-2 (in PFU/mL) = Average number of plaques in a particular viral dilution ÷ (dilution factor × 0.2 mL).

### H$_2$S detection assays

H$_2$S generation was measured using methylene blue assay [25] and lead acetate assay [19]. For methylene blue assay, the supernatant of cells treated with NaHS or GYY4137 was incubated with zinc acetate (1%) and NaOH (3%) (1:1 ratio) to trap H$_2$S for 30 min. The reaction was terminated using 10% trichloroacetic acid solution. Following this, reactants were incubated with 20 mM N,N-dimethylphenylendiamine (NNDPD; Sigma-Aldrich) in 7.2 N HCl and 30 mM FeCl$_3$ in 1.2 N HCl

for 30 min, and absorbance was measured at 670 nm. The concentration of $H_2S$ was determined by plotting absorbance on a standard curve generated using NaHS (0–400 µM; $R^2 = 0.9982$). For lead acetate assay, lead acetate papers were prepared by soaking Whatman filter paper in 5 mM lead acetate solution for at-least 2 h and then air dried. Cell lysate, prepared in passive lysis buffer, was incubated with 10 mM cysteine (Sigma-Aldrich) and 1 mM PLP (Sigma-Aldrich) in a 96-well plate. The lid of the plate was covered with lead acetate paper, and incubated at 37°C overnight. Intensity of the brown precipitates was calculated using Image J software.

### Flow cytometry and redox potential measurements

Flow cytometer was conducted to obtain the ratio metric response of cells transfected with Grx1-roGFP2 sensor plasmids, as described in our earlier studies [66,68]. Biosensor-expressing cells infected/uninfected were harvested after treating with 1 mM NEM and fixed with 4% PFA before analyzing in BD FACS Verse. For each experiment, the minimal and maximal fluorescence ratios corresponding to 100% sensor reduction and 100% sensor oxidation was calculated using DTT (10 mM) as the reducing agent and $H_2O_2$ (10 mM) as the oxidizing agent. The observed ratios were used to determine the degree of biosensor oxidation.

### Measurement of oxygen consumption rates

OCRs were measured using a Seahorse XFp extracellular flux analyzer (Agilent Technologies) as per the manufacturer's instructions. Briefly, cells (VeroE6 cells infected with SARS-CoV-2 in presence or absence of 10 µM GYY4137) were seeded at a density of $10^4$–$10^5$ per well in a Seahorse flux analyzer plate precoated with Cell-Tak (Corning). Cells were incubated for 1 h in a non-$CO_2$ incubator at 37°C before loading the plate in the Seahorse analyzer. To assess mitochondrial respiration, three OCR measurements were performed without an inhibitor in XF assay media to measure basal respiration, followed by sequential addition of oligomycin (1 µM), an ATP synthase inhibitor (complex V). Three OCR measurements were made to determine ATP-linked OCR and proton leakage. Next, cyanide-4-(trifluoromethoxy) phenylhydrazone (FCCP; 0.25 µM), was injected to determine the maximal respiration rate and the spare respiratory capacity. Finally, rotenone (0.5 µM) and antimycin A (0.5 µM), inhibitors of NADH dehydrogenase (complex I) and cytochrome *c* - oxidoreductase (complex III), respectively, were injected to completely shut down the ETC to determine non mitochondrial OCR (nmOCR). Mitochondrial respiration parameters were analyzed using Wave Desktop 2.6 software (Agilent Technologies).

### Protein persulfide detection by ProPerDP

Protein persulfidation was detected by the ProPerDP method established by Doka et al [21]. Briefly, VeroE6 cells, either uninfected or infected with 0.1 MOI SARS-CoV-2 for 24 h, were rinsed with phosphate-buffered saline (PBS) and incubated with 2 µM Auranofin, followed by incubation with 1 mM IAB (ThermoFisher Scientific, USA) in Hank's balanced salt solution (HBSS) for 3 h at 37°C. The cells were then washed two times with HBSS and lysed by scraping in lysis buffer [40 mM HEPES, 50 mM NaCl, 1 mM EGTA, 1 mM EDTA (pH 7.4), and 1% CHAPS] containing 1% protease inhibitor cocktail (Sigma-Aldrich). Cell supernatant was clarified by centrifugation at 15,000 x g for 15 min at 4°C. Protein concentration was determined using the Bradford assay. Biotinylated proteins were pulled down by streptavidin-coated magnetic beads (Invitrogen) for 2 h with rotation. After incubation, magnetic beads were separated from the solution phase with a magnetic particle separator. The supernatant was placed in a clean tube, and the beads were washed three times with Tris-buffered saline containing 0.05% Tween 20 (TBST). The beads were then resuspended in 25 mM DTT and incubated for 30 min with gentle mixing. The magnetic separation was repeated. The supernatant containing persulfidated proteins and the beads were finally boiled at 100°C for 3 min in SDS-PAGE sample loading dye. The samples were then analysed by SDS-PAGE gel electrophoresis, followed by staining with colloidal Coomassie stain to visualize the bands.

## RNA sequencing

Total RNA was extracted from VeroE6 cells infected with 0.01 SARS-CoV-2 in the presence or absence of 5 mM GYY4137. Following extraction, the RNA was quantified and assessed for purity by a 2100 Bioanalyzer (Agilent Technologies, Waldbronn, Germany). RNA samples with an RIN (RNA Integrity Number) value > 8 were processed further for sequencing. mRNA enrichment was performed using NEB mRNA enrichment kit as per manufacturer's protocol, and the concentration of enriched mRNA was quantified by Qubit RNA HS Assay Kit (Life Technologies, USA). Libraries were prepared using NEB Next Ultra Directional RNA Library Prep Kit for Illumina (New England Biolabs, USA), according to manufacturer's instructions. The library size distribution and quality were assessed using a high sensitivity DNA Chip (Agilent Technologies, USA) and sequenced in NovaSeq 6000 (Illumina, USA) sequencer using 1X50 bp single-end reads with 1% PhiX spike-in control.

## Differential gene expression and statistical analysis for RNA-Seq

Raw reads were obtained as fastq files. The reference genome sequence and annotation files for *Chlorocebus sabaeus* (GCF_015252025.1_Vero_WHO_p1.0) were downloaded from the NCBI ftp ("ftp.ncbi.nlm.nih.gov"). The raw read quality was checked using the FastQC software (version v0.11.5)[111]. Differential gene expression (DGE) analysis was adapted from Pertea et.al, 2016. HISAT2 (v2.2.1) was used to index the reference genome and align the raw reads on it [112,113]. The resulting sam files were sorted and converted to bam files using SAMTOOLS (v1.15.1) [114]. A matrix of raw read count per gene was generated using stringtie2 (v.2.2.1) package [115]. Genes which were annotated in the reference genome and had atleast 10 reads in minimum of two replicates of any samples, were considered for further analyses. For DGE analysis, the methodology in Chen Y et. al, 2016 was performed using edgeR package (v4.0.2) [116,117]. In each comparison, the absolute fold change (FC) of 1.5 and false discovery rate (FDR) of 0.1 was taken as the threshold for defining the genes with differential gene expression. fgsea package (v1.28.0) was used to perform pathway enrichment anaylsis resulting in normalised enrichment score [118].

## Metabolite extraction and analysis

Untargeted metabolomics analysis was performed using Q Exactive Hybrid Quadrupole Orbitrap high-resolution mass spectrometer with an ESI source (ThermoFisher Scientific, Inc., USA). Briefly, 1 x $10^6$ VeroE6 cells were infected with SARS-CoV-2 (MOI-0.1) in the presence or absence of 10 μM GYY4137. Cells were incubated for 24 h before harvesting in extraction buffer. The cells were extracted quickly using extraction buffer (methanol/acetonitrile/water (50:30:20 v/v/v)) at -20°C on dry ice by scrapping. Extracted metabolite solution was kept for 2 h at -80°C followed by centrifugation twice at 20,000 x g for 20 mins at 4°C. Finally, the supernatant was vacuum dried using a speed vac. Before injecting the samples for LC/MS, lyophilized metabolites were dissolved in 30 μL of 50% acetonitrile and 0.1% formic acid. Further, the samples were vortexed and centrifuged at 12,000 x g for 15 mins. Supernatant was collected, and 5 μL/replicate was injected into the instrument. Samples were run in triplicate using electrospray ionization (ESI)-positive mode. Solvent (A) contained water and 0.1% formic acid, and Solvent (B) contained methanol and 0.1% formic acid. The capillary temperature was set to 320°C. The sheath gas flow rate was set to 60, the aux gas flow rate to 20, and the spray voltage to 3.5kV. The flow rate of the LC-MS instrument was set to 0.500 mL/min. All method files were written and executed via Thermo Xcalibur 4.0 software (ThermoFisher Scientific, Inc., USA). The raw data obtained were processed using compound discoverer 2.0 software and Metaboanalyst 6.0[119].

## Animal experiments

***Ethics and animals' husbandry*** The Institute Biosafety Committee (IBSC) and the Institute Animal Ethical Committee (IAEC) evaluated and approved the work plans for the animal experiments, and the experiment was carried out in

accordance with CPCSEA criteria. Ten-to-twelve-week-old female Balb/c mice were used in the animal experiment, along with the required numbers of gold Syrian hamsters (*Mesorectums auratus)* of both sexes weighing between 50–60 grams. All experiments were performed inside the virus BSL-3 laboratory at the Indian Institute of Science, Bengaluru, India. Animals used in the experiments were kept in individually ventilated cages (IVCs) with access to pellet feed, water *ad libitum*, and a 12-h day/night light cycle. Furthermore, the temperature and relative humidity of the viral BSL-3 laboratory were kept at $23 \pm 1$ ◦C and $50 \pm 5\%$, respectively.

***Virus infection and treatment experiments*** After acclimatization for seven days in IVC cages in the virus BSL-3 laboratory, the experimental animals (mice and hamsters) were randomly grouped for drug treatment, vehicle control, and uninfected groups (n > 4 (for hamster); n > 8 (for mice)). Animals were sedated and anesthetized using a cocktail of ketamine (90 mg/kg for mice and 150 mg/kg for hamsters) and xylazine (4.5 mg/kg for mice and 10 mg/kg for hamsters) intraperitoneally, and they were intranasally infected with $10^5$ PFU SARS-CoV-2 (US isolate) in 100 µL PBS (for hamsters) or $5 \times 10^4$ PFU SARS-CoV-2 (MA-10 isolate) in 50 µL PBS (for mice). Treatment involved intranasal administration of 50 mg/kg.b.wt. of GYY4137 or Na-GYY4137 1 h before infection and 6 h and 24 h post infection. Body weight was recorded each day during the course of the experiment until the animals were sacrificed at either 3 or 5 dpi (for mice) or 4 dpi (for hamsters). At the respective time points of the experiment, all animals (mice and hamsters) were euthanized through intraperitoneal injection of an overdose of Ketamine (Bharat Parenteral Limited) and Xylazine (21, Indian Immunological Ltd) cocktail. The left lobe of lung was harvested and fixed in 4% paraformaldehyde (PFA) for histopathological examination of lungs. The right lobes were frozen at -80ºC for determining the virus copy number using RT-qPCR or plaque assay. Briefly RNA was isolated using TRIzol method. Equal concentration of RNA was used for viral titre estimation using Q-line (nCoV-19) RT-PCR detection kit (Q-line Biotech private limited), as per manufacturer's protocol. Standard curve with known viral titre was generated to calculate the viral e gene copy number.

***Histopathological Examination*** Paraformaldehyde-fixed lungs were processed, embedded in paraffin, and cut into 4 µm sections by microtome for haematoxylin and eosin staining. The lung sections were microscopically examined and evaluated for different pathological scores. For lung tissue histopathology scoring, we developed a method using Mitchison's virulence scoring system with modification, considering the consolidation of lungs, severity of bronchial and alveolar inflammation, immune cell influx, and alveolar and perivascular edema [120,121]. The histopathology scores were graded as 0–4 (4: severe pathology; 3: moderate pathology; 2: mild pathology; 1: minor/minimum pathology; 0: no pathology).

***Whole Body Plethysmography*** For whole body plethysmography (WBP), 18 week old female Balb/c mice were infected with $10^5$ PFU of mouse-adapted SARS-CoV-2 (MA10) virus under xylazine-ketamine anesthesia. One of the infected groups received 50 mg/kg Na-GYY4137 intranasally 1 h before, and 6 h, and 24 h after infection. The other infected group was intranasally dosed with the same volume of PBS at specific time points. A second dose of infection was given to mice in the two infected groups 48 h after the first dose. All the mice were subjected to respiratory function evaluation before infection and every day post-infection for 7 days. Mice were allowed to acclimatize in the subject chambers of the whole-body plethysmograph (Vivoflow, Scireq), and respiratory function parameters were recorded for 5 minutes each day using iox software (v2.10.8.38). The recordings were captured at every 15-second interval, and data were analyzed using two-way ANOVA.

## Statistical analysis

All statistical analyses were performed using GraphPad Prism software for Macintosh (version 9.5.0). The data values are indicated as mean ± S.D. For statistical analysis, Student's *t*-test (in which two groups are compared) and one-way or two-way ANOVA (for analysis involving multiple groups), with appropriate corrections were used.

## Supporting information

**S1 Fig.** **(A)** Time-dependent changes in expression of *cbs*, *mst* and *cth* during SARS-CoV-2 (HK variant) replication in Calu-3 cells by RT-qPCR. **(B)** Time-dependent changes in expression of *cbs*, *mst* and *cth* during SARS-CoV-2-HK replication in HEK-ACE2 cells by RT-qPCR. **(C)** Protein levels of CBS, CTH and MST during SARS-CoV-2-HK replication in HEK-ACE2 cells, quantified by densitometric analysis using Image Lab software.
(TIF)

**S2 Fig.** **(A)** Knockdown confirmation of CBS in VeroE6 cells by western blotting. **(B)** Knockdown confirmation of *mst* in VeroE6 cells by RT-qPCR. **(C)** SARS-CoV-2 viral load in knockdown VeroE6 cells (ShCBS/CTH/MST). **(D)** Viability of VeroE6 cells in the presence of GYY4137 and Na-GYY4137 at 48 h post treatment by MTT assay. **(E)** Plaque assay from culture supernatant of different treatment groups. **(F)** *ace2* expression upon SARS-CoV-2 infection and GYY4137 treatment in VeroE6 cells. **(G)** *ace2* and *tmprss2* expression in Calu-3 cells, infected with 0.1 MOI SARS-CoV-2 in presence or absence of 5 mM GYY4137 at 48 h p.i. **(H)** Time of addition experiment of drug GYY4137 in VeroE6 cells.
(TIF)

**S3 Fig.** **(A)** *nfe2l2* transcript counts upon SARS-CoV-2 infection in presence or absence of GYY4137 by RNA sequencing. **(B)** Heat maps of genes associated with oxidoreductase activity.
(TIF)

**S4 Fig.** **(A)** RT-qPCR analysis of Nrf2 regulated genes in Vero-shKeap1 cells.
(TIF)

**S5 Fig.** **(A)** Viral load in SARS-CoV-2 infected mice in presence or absence of Na-GYY4137 measured by RT-qPCR at 7 day p.i. **(B)** Lung function parameters measured by whole body plethysmography.
(TIF)

**S1 Table.** **List of shRNA constructs used in the study.**
(DOCX)

**S2 Table.** **List of primers used in the study.**
(DOCX)

**S1 Data.** **Normalized CPM counts (transcriptomics data).**
(XLSX)

## Acknowledgments

We thank Prof. S. Vijaya (IISc), Prof. Varadharajan Sundaramurthy (NCBS) and Dr. Shashank Tripathi (IISc), for providing cell lines and virus stocks. We also thank Prof. Deepak Kumar Saini (IISc) for providing the shRNA constructs. We are thankful to Dr. Awadesh Pandit and Next Generation Genomics Facility (NGGF) at the National Centre for Biological Sciences (NCBS), Bengaluru for conducting the RNA-seq experiment. We acknowledge the viral biosafety level 3 (Viral BSL-3) and COVID-19 diagnostic facilities at CIDR, IISc, Bengaluru. We thank Karl Drlica for providing critical comments on the manuscript.

## Author contributions

**Conceptualization:** Ragini Agrawal, Amit Singh.

**Data curation:** Ragini Agrawal.

**Formal analysis:** Ragini Agrawal, Virender Kumar Pal, Suhas KS, Gopika Jayan Menon, Inder Raj Singh, Nitish Malhotra, Naren CS, Kailash Ganesh, Raju S Rajmani, Aswin Sai Seshasayee, Nagasuma Chandra.

**Funding acquisition:** Amit Singh.

**Investigation:** Ragini Agrawal.

**Methodology:** Ragini Agrawal, Virender Kumar Pal, Suhas KS, Gopika Jayan Menon, Raju S Rajmani.

**Project administration:** Ragini Agrawal, Amit Singh.

**Resources:** Amit Singh.

**Software:** Inder Raj Singh, Nitish Malhotra, Naren CS, Kailash Ganesh, Aswin Sai Seshasayee, Nagasuma Chandra, Manjunath B Joshi.

**Supervision:** Manjunath B Joshi, Amit Singh.

**Validation:** Ragini Agrawal, Amit Singh.

**Writing – original draft:** Ragini Agrawal, Amit Singh.

**Writing – review & editing:** Ragini Agrawal, Virender Kumar Pal, Suhas KS, Gopika Jayan Menon, Inder Raj Singh, Nitish Malhotra, Naren CS, Kailash Ganesh, Raju S Rajmani, Aswin Sai Seshasayee, Nagasuma Chandra, Manjunath B Joshi, Amit Singh.

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
