## [Decision Letter · Decision Letter 0]

18 Feb 2025

PPATHOGENS-D-25-00105

Hydrogen sulfide (H2S) coordinates redox balance, carbon metabolism, and mitochondrial bioenergetics to suppress SARS-CoV-2 infection.

PLOS Pathogens

Dear Dr. Singh,

Thank you for submitting your manuscript to PLOS Pathogens. After careful consideration, we feel that it has merit but does not fully meet PLOS Pathogens's publication criteria as it currently stands. Therefore, we invite you to submit a revised version of the manuscript that addresses the points raised during the review process.

Please submit your revised manuscript within 60 days Apr 19 2025 11:59PM. If you will need more time than this to complete your revisions, please reply to this message or contact the journal office at plospathogens@plos.org. Please include the following items when submitting your revised manuscript:

We look forward to receiving your revised manuscript.

Kind regards,

Yong-Hui Zheng, Ph.D.

Guest Editor

PLOS Pathogens

Michael Letko

Section Editor

PLOS Pathogens

 Sumita Bhaduri-McIntosh

Editor-in-Chief

PLOS Pathogens

orcid.org/0000-0003-2946-9497

Michael Malim

Editor-in-Chief

PLOS Pathogens

orcid.org/0000-0002-7699-2064

**Additional Editor Comments :**

Please revise your manuscript based on the two reviewers' comments. Please keep in mind that additional experiments must be conducted to address their concerns.

**Journal Requirements:**

At this stage, the following Authors/Authors require contributions: Ragini Agrawal, Virender Pal, Suhas KS, Gopika Menon, Inder Raj Singh, Nitish Malhotra, Naren CS, Kailash Ganesh, Raju Rajmani, Aswin Seshasayee, Nagasuma Chandra, Manjunath Joshi, and Amit Singh. Please ensure that the full contributions of each author are acknowledged in the "Add/Edit/Remove Authors" section of our submission form.

- ® on page: 33

- TM on page: 24.

4) Thank you for including an Ethics Statement for your study. Please include:

i) A statement that formal consent was obtained (must state whether verbal/written) OR the reason consent was not obtained (e.g. anonymity). NOTE: If child participants, the statement must declare that formal consent was obtained from the parent/guardian.].

5) Please upload all main figures as separate Figure files in .tif or .eps format. For more information about how to convert and format your figure files please see our guidelines: 

Potential Copyright Issues:

i) Figures 1A, 4A, 5A, 5F, 6A, and 6F. Please confirm whether you drew the images / clip-art within the figure panels by hand. If you did not draw the images, please provide (a) a link to the source of the images or icons and their license / terms of use; or (b) written permission from the copyright holder to publish the images or icons under our CC BY 4.0 license. Alternatively, you may replace the images with open source alternatives. See these open source resources you may use to replace images / clip-art:

7) Thank you for stating that "The RNA-seq datasets have been submitted in Gene Expression Omnibus (GEO) with accession number GSE283665." We strongly recommend all authors deposit their data before acceptance, as the process can be lengthy and hold up publication timelines. Please note that, though access restrictions are acceptable now, your entire minimal dataset will need to be made freely accessible if your manuscript is accepted for publication. This policy applies to all data except where public deposition would breach compliance with the protocol approved by your research ethics board. If you are unable to adhere to our open data policy, please kindly revise your statement to explain your reasoning and we will seek the editor's input on an exemption.

8) Please ensure that the funders and grant numbers match between the Financial Disclosure field and the Funding Information tab in your submission form. Note that the funders must be provided in the same order in both places as well. Currently, the order of the grants is different in both places. In addition, " Crypto Relief fund (ODAA/INT/20-21) and DST-FIST for infrastructure support to IISc" are missing from the Funding Information tab.

**Reviewers' Comments:**

Reviewer's Responses to Questions

**Part I - Summary**

Reviewer #1: Ragini Agrawal et al. demonstrate that H2S gas participates in the modulation of SARS-CoV-2 infection by suppressing virus replication. The down-regulation of H2S-producing enzymes, including CBS, CTH, and 3-MST, has been observed to be associated with virus replication. The use of GYY4137, a slow-releasing H2S donor, for chemical supplementation of H2S was found to diminish virus replication. This was achieved by inducing the Nrf2/Keap1 pathway, restoring redox balance and carbon metabolites, and potentiating mitochondrial oxidative phosphorylation.

This study is well-written and nicely organized and presented, with carefully crafted figures. However, I believe there are a number of minor to moderate changes that could be introduced to improve the manuscript.

Reviewer #2: In this manuscript, the authors investigate the role of H2S on SARS-CoV-2 replication. Previous research from this group and others has demonstrated the antiviral properties of H2S on respiratory pathogens. The authors conclude that SARS-CoV-2 limits intracellular H2S in infected cells and H2S inhibits viral replication through modulation of the metabolic pathways. Further increasing H2S through pharmalogical intervention blunted viral replication in the lung of infected mice and hamsters, leading to reduced pathology and respiratory distress.

This is a very well written manuscript with several different lines of evidence that support the authors conclusion. I have noted below one alternative hypothesis that should be addressed.

**Part II – Major Issues: Key Experiments Required for Acceptance**

Reviewer #1: 1.Previous papers (PMID: 32515982) have shown that H2S may block SARS-CoV-2 entry into host cells by interfering with ACE2 and TMPRSS2, inhibit SARS-CoV-2 replication by attenuating virus assembly/release, and protect SARS-CoV-2-induced lung damage by suppressing immune response and inflammation development. I suggest the authors to detect the expression of ACE2 and TMPRSS2 in the paper to exclude its function in the decreased virus replication.

2.In Fig 5I: “We noticed that treatment with GYY4137 specifically reversed the effect of SARS-CoV-2-HK infection on the expression of genes associated with the oxidoreductase complex and Nrf2-antioxidant pathway (Fig. 3I)”. Were the oxidoreductase complex also analyzed in the transcriptional data?

3.The full dataset of the RNA-seq needs to be provided in the supplemental information.

4.In line 247-250: “Consistent with this finding, we observed down-regulation of antioxidant genes (e.g., glutathione peroxidases [gpx4], peroxiredoxins [prdx6], hemoxygenase (hmox1], catalase [cat], and thioredoxins [txnrd1])) regulated by nuclear factor erythroid 2-related factor 2 (Nrf2)-a central controller of cellular resistance to redox stress[34]”. This concluded from the experiment or Reference 34? In addition, I could find prdx6 in the results throughout the manuscript.

5.In line 255-257: “As with our RT-qPCR data, infection with SARS-CoV-2 reduces the expression of genes associated with sulfur metabolism, which include H2S biogenesis (Fig. 3D).” The results presented in Figure 3D are inconsistent with the description.

6.The authors should clarify why the Nrf2 gene was not included in the RNA-seq data, despite its expression being detected by WB.

7.According to the results, H2S suppression of SARS-CoV-2 proliferation is associated with the expression of the Nrf2/Keap-1 regulon, redox metabolites, and mitochondrial function. Therefore, what is the relationship among these factors? Which one serves as the primary influencing factor?

8.The authors should clarify the rationale behind using different treatments in Figure 3 and Figure 4. Additionally, they should discuss whether this discrepancy in treatment affects the results obtained.

Reviewer #2: The authors conclude that H2S suppresses viral replication through modulation of host cell metabolic pathways. However an alternative conclusion from the data could be that H2S inhibits viral entry and cells are less infected. All of the experiments in this manuscript pretreat the cells/mice with higher levels of H2S, either through treatment with a slow H2S release chemical or genetic manipulation of the H2S synthesis pathways. To address this alternative hypothesis, I suggest the following experiment. Infect cells at a high MOI (3-5). Wash cells after one hour and treat with H2S. Monitor viral growth kinetics.

**Part III – Minor Issues: Editorial and Data Presentation Modifications**

Reviewer #1: 1.Fig. 2A: The label for Fig. 2A is missing in the Results section.

2.In line 225-232: I was unable to locate the genes in Fig 3, and the reference was not provided either. Kindly provide a detailed description of it.

3.Fig. 6I: Based on the text, there should be a label (7) missing in Fig. 6I. Furthermore, please clarify the meaning of 'BV' in Fig. 6I?

4.In lines 1314-1315: “(B) denotes bronchial lumen and (AS) denotes alveolar space”. These annotations may require revision.

Reviewer #2: - Fig 1H. Violin plots are distracting. Please show scatter plot of data points.

- Fig 6B right panel. Y-axis title should read n gene copy number. Also how is this data normalized?

- Fig 6 general: Make groups across different panels the same color. For example SARS-CoV-2 + GYY4137 is red in 6A-E, then green in 6F-H.

PLOS authors have the option to publish the peer review history of their article (what does this mean? ). If published, this will include your full peer review and any attached files.

**Do you want your identity to be public for this peer review?** For information about this choice, including consent withdrawal, please see our Privacy Policy .

Reviewer #1: No

Reviewer #2: No

**Figure resubmission:**
---

## [Decision Letter · Decision Letter 1]

28 Apr 2025

Dear Dr Singh,

We are pleased to inform you that your manuscript 'Hydrogen sulfide (H2S) coordinates redox balance, carbon metabolism, and mitochondrial bioenergetics to suppress SARS-CoV-2 infection.' has been provisionally accepted for publication in PLOS Pathogens.

Best regards,

Yong-Hui Zheng, Ph.D.

Guest Editor

PLOS Pathogens

Michael Letko

Section Editor

PLOS Pathogens

Sumita Bhaduri-McIntosh

Editor-in-Chief

PLOS Pathogens

orcid.org/0000-0003-2946-9497

Michael Malim

Editor-in-Chief

PLOS Pathogens

orcid.org/0000-0002-7699-2064

Reviewer Comments (if any, and for reference):

Reviewer's Responses to Questions

**Part I - Summary**

Reviewer #1: The authors have adequately addressed the concerns and suggestions raised in the initial review. The revisions have improved the manuscript's clarity, rigor, and overall quality. The study provides valuable insights into the role of H₂S in modulating SARS-CoV-2 infection, particularly in suppressing viral replication through the Nrf2/Keap1 pathway, redox balance restoration, and mitochondrial oxidative phosphorylation. This study makes a meaningful contribution to understanding host-directed antiviral strategies and highlights the therapeutic potential of H₂S donors in SARS-CoV-2 infection.

The manuscript remains well-organized, with logical flow and carefully designed figures. Since the authors have satisfactorily addressed all previous concerns, I recommend accepting it in its current form. No further revisions are required.

This review acknowledges the improvements made and confirms that the manuscript is now suitable for publication.

Reviewer #2: Thank you for addressing all critiques.

**Part II – Major Issues: Key Experiments Required for Acceptance**

Reviewer #1: (No Response)

Reviewer #2: N/A

**Part III – Minor Issues: Editorial and Data Presentation Modifications**

Reviewer #1: (No Response)

Reviewer #2: N/A

---

## [Editor Report · Acceptance letter]

Dear Dr Singh,

We are delighted to inform you that your manuscript, "Hydrogen sulfide (H2S) coordinates redox balance, carbon metabolism, and mitochondrial bioenergetics to suppress SARS-CoV-2 infection.," has been formally accepted for publication in PLOS Pathogens.

Best regards,

Sumita Bhaduri-McIntosh

Editor-in-Chief

PLOS Pathogens

orcid.org/0000-0003-2946-9497

Michael Malim

Editor-in-Chief

PLOS Pathogens

orcid.org/0000-0002-7699-2064